# Delivery of Therapeutic Biopolymers Employing Silica-Based Nanosystems

**DOI:** 10.3390/pharmaceutics15020351

**Published:** 2023-01-20

**Authors:** Zoila Gándara, Noelia Rubio, Rafael R. Castillo

**Affiliations:** 1Departamento de Química Orgánica y Química Inorgánica, Universidad de Alcalá, 28805 Alcalá de Henares, Spain; 2Instituto de Investigación Química “Andrés M. del Río” (IQAR), Universidad de Alcalá, 28805 Alcalá de Henares, Spain

**Keywords:** mesoporous silica nanoparticles, biopolymers, nanomedicine, nanotechnology

## Abstract

The use of nanoparticles is crucial for the development of a new generation of nanodevices for clinical applications. Silica-based nanoparticles can be tailored with a wide range of functional biopolymers with unique physicochemical properties thus providing several advantages: (1) limitation of interparticle interaction, (2) preservation of cargo and particle integrity, (3) reduction of immune response, (4) additional therapeutic effects and (5) cell targeting. Therefore, the engineering of advanced functional coatings is of utmost importance to enhance the biocompatibility of existing biomaterials. Herein we will focus on the most recent advances reported on the delivery and therapeutic use of silica-based nanoparticles containing biopolymers (proteins, nucleotides, and polysaccharides) with proven biological effects.

## 1. Introduction

Biomacromolecules play a fundamental role in many biological aspects. For example, they are responsible for all signaling and metabolic processes, thus contributing to the structural integrity of eukaryotic and prokaryotic cells or encoding genetic information, among many other roles. Therefore, the use of biomacromolecules in advanced new-generation therapies is one of the most promising fields of research. For example, regarding nucleic acids, the transfection strategies that allowed to genetically modify unicellular organisms to produce human enzymes and hormones have been clearly outranged by current technology. Recently, the milestone of gene editing on multicellular organisms has also been achieved, opening the way to more complex therapies [1]. In addition to that, chemical modifications of nucleic acids have also arisen as a potent strategy to develop new generation biomedicines [2,3]. On the other hand, proteins and peptides have also contributed to the development of relevant immunotherapeutics [4,5], immunodetection methods, and to the treatment of chronic diseases such as diabetes, growth disorders, cancer [6] and new-generation antibacterial therapies [7,8]. In all these therapies, biomacromolecules with unique 3D structures play a crucial role in tuning biological processes in a way that typical small molecules and drugs are not able to exert. However, the use of those macromolecular biotherapeutics is extremely challenging, as they must penetrate across biological membranes or physiological barriers that they are not able to in order to perform their therapeutic action [9]. This is the case of the messenger RNA molecules of the anti-COVID Comirnaty^®^ and Spikevax^®^ vaccines, which require two protective strategies to carry out their function. In both cases, a liposomal coating is used to preserve the integrity of the mRNA during trafficking in the bloodstream. However, this modification alone is not enough to achieve a suitable effect and uracil must be replaced by pseudouridine within the sequence in order to intracellularly increase mRNA lifespan and thus produce enough amounts of viral protein to provoke immune responses [10,11].

Regarding cancer treatment, the discovery of the enhanced permeation and retention (EPR) effect [12] in solid tumors has made it possible to create a huge number of possible biomedical solutions. This is a consequence of how nanoparticles interact with tissues and living cells, which typically differ from common small molecule–cell interactions. However, since most of these nanoparticles are not of biogenic origin, there is a major limitation to their widespread use, especially when these systems are dosed intravenously or parenterally where immune responses are typically triggered [13]. Nevertheless, current nanotechnology also offers significant advances in immune stealthing materials [14,15], where polyethylene glycol outranges any other material. Nevertheless, some current developments in the field of biopolymers are also able to accomplish this task, but with additional functional features, which is of enormous interest for the development of therapeutic nanohybrids.

Regarding nanoparticles, there are an overwhelming number of them. They can be classified in many ways on the basis of their different chemical nature, shape, size, and topology [16,17]. However, among them, liposomes display the highest loading capacity and the most advanced clinical development [18], although they also present several limitations. For example, the poor thermal, mechanical, and chemical stability of liposomes complicates the development of multifunctional liposomal systems and the development of formulations for co-delivery, for which reason there is a need to improve the current drug delivery technology. Among all available nanosystems, porous silica is one of the most widely employed for the development of potential nanotherapeutics, even though mesoporous silica is still not approved for use in clinical practice [19,20]. Nevertheless, SiO_2_ is generally considered a safe material by the FDA and is widely employed in food industry and even as a structural basis for certain living organisms [21]. Indeed, there are many examples in which biosafety and biocompatibility of silicas have been demonstrated [22,23,24], suggesting that silica may be used either alone or as a component for the development of new nanotherapeutics, being even employed by certain diatom algae as structural materials [21]. In this review, the most relevant advances based on porous silica particles able to deliver biomacromolecules with therapeutic potential are covered, as well as other targeting strategies [25,26,27] and those employed as diffusion barriers in the development of controlled release nanosystems [28,29] (Figure 1).

## 2. Development of Silica-Based Nanomedicines Using Therapeutic Nucleotides

The huge potential of nucleotides for developing advanced therapies has resulted in a very fruitful field of research. Among the most relevant therapeutic applications, there could be highlighted the following: (1) gene transfection with plasmids [30,31], (2) development of vaccines and protein modulation therapies employing miRNAs [32,33], (3) gene silencing therapies [34,35], and (4) the most recent to arrive, CRISPR genome editing [36,37]. However, like with many other therapeutic biomolecules, the indiscriminate use of nucleotides is not easy and immediate, as they suffer from quick degradation in extracellular regions and intracellular environments and are not able to cross biological membranes. For such reasons, DNA and RNA-based biotechnology must be also accompanied by a development on delivery technologies. Historically, viral-based vectors allowed for development of transfection processes successfully, but the difficulty of handling and the triggering of immune responses, discouraged from their use. This has forced the scientific community to develop other non-immunogenic systems able to deliver these therapeutic biopolymers to both cytoplasm and cell nucleus. Among them, lipofectamine technology makes it possible to efficiently transfect plasmids in vitro but not in vivo; therefore, new systems need to be developed.

Chromosomes, based on DNA sequences, are responsible for keeping the genetic information. For such reason living cells have established biological mechanisms to preserve them unaltered within the nucleus but to destroy them in cytoplasmatic regions. On the other hand, RNA nucleotides responsible for encoding protein expression must be also removed from the cytoplasm once their message is delivered, as otherwise they may induce homeostasis imbalances due to excessive protein expression. The consequence is that both species are quickly metabolized by living organisms to maintain homeostasis. For this reason, it is important to create carriers able to protect these biopolymers when dosed in vivo. This problem has been addressed for more than a decade since the pioneering work of Lin [38] who opened the way to the use of mesoporous silica systems for nucleotide delivery. This topic was visited 5 years ago in a previous contribution by some of us; in that review [39], the known strategies for the preparation of DNA/RNA-based silica nanosystems were detailed [39]. Despite that review covering all relevant aspects of the development of nucleotide-based nanomedicines, we revisit them herein, focusing on the most recent advances (Table 1). Nevertheless, as this review is focused on the therapeutic profile of biopolymers, cell targeting [29,40,41,42,43,44], biosensors [45] and pore gating [46,47] strategies [29,40,41,42,43,44,45,46,47] will be omitted.

As highlighted elsewhere [39,48], effective transfection is mainly achieved in the presence of polycationic species, which upon electrostatic interaction with negatively charged DNA/RNA provide a complex structure. If this structure is still able to interact with the negatively charged membrane, it may favor cell uptake and deliver the nucleotide intracellularly. The main problem with this technology is the membrane-lytic effect caused by polycations; which could be partially avoided by using nanoparticles as nonviral vectors. In addition to the more convenient nanoparticulate formulation, the use of more benign biogenic polymers, i.e., polylysine (PLL) instead polyamidoamine (PAMAM) dendrimers or polyethyleneimine (PEI) [49], have significantly improved the applicability of these vectors.

In general, polycationic MSNs show lower toxicity levels than free polymers. This effect could be understood considering two different but complementary aspects: (1) the polycationic particle has a diffuse distribution of positive charges along the surface and hence lower destructive interactions with membranes and (2) when building multilayered assemblies, polycationic components are partially balanced with the negatively charged silica, thus reducing the overall toxicity by lowering the number of cationic groups. Both approaches have been successfully employed for the construction of transfection agents as previously reported [39]. Reported strategies for nucleotide delivery are summarized in Figure 2 and Table 1.

Focusing on nanoparticle morphology, Wang et al. demonstrated that small (ca. 50 nm) dendritic mesoporous silica nanoparticles with large pores (>20 nm) were highly effective transfection agents when coated with 10 KDa PEI, as demonstrated with the green fluorescent protein plasmids [50]. In that work, the authors claimed that the dendritic morphology provided better transfection profiles than regular MSNs (up to ca. 40%). This effect, although not detailed in depth, could be a consequence of the better hosting and protection of plasmids at the larger mesopores. In another contribution, Su, Deng and coworkers, studied the performance of ultrasmall MSNs for siRNA delivery employing PEI too [51]. Their nanosystem, with an overall size of less than 12 nm, showed good biocompatibility on cancerous HeLa cultures (up to 200 μg·mL^−1^) and up to 80% knockdown onto glyceraldehyde-3-phosphate dehydrogenase (GADPH) onto HEK-293T cells, suggesting a great transfection efficiency similar to the reference, Lipofectamine 2000. This system employed a 1800 Da PEI chemically grafted onto the surface of modified aminopropyl triethoxysilane (APTES) together with a short, crosslinked polyethylene glycol (PEG) chain. This approach, based on the post-functionalization of MSNs with APTES, made it possible to significantly reduce the overall number of grafted PEI chains and thus to reduce the number of positive charges and hence, overall toxicity.

The modification of silica’s textural properties aimed at maximizing the loading efficiency of DNA is also an interesting aspect. Along this line, an article by Yu, Song, and coworkers studied with the binding behavior of PEI and DNA onto phosphonate-modified silica nanoparticles with different surface roughness [52]. Herein, the authors prepared different asymmetric nanoparticles with both smooth and spiky hemispheres within the same nanoparticle. According to their investigations, the larger spiky surface the better the deposition of PEI-DNA. In addition to the better loading efficiency, larger spiky surfaces also provided lower hemolysis rates. As a result, the nanoparticle with ca. 93% spiky surface showed the best behavior with a loading rate of up to 100 μg·mL^−1^ and a red blood cell hemolysis rate of ca. 40% for 1200 μg of MSNs per mL concentration. These asymmetric particles were evaluated on the transfection of GFP in HEK-293T, reaching a 35% of effective transfection, which suggests that this kind of structured surfaces could help on the rational design of more biocompatible and efficient nonviral vectors. Perhaps, although not studied, the phosphonate groups present at the silica surface may be responsible for such good performance because of their ubiquity on cell membranes and biomolecules and its highly convenient interaction both in vitro and in vivo.

Despite the advances achieved by tuning the morphological properties of silica, it is also important to consider the chemical nature of PEI, which may also exert an additional toxic effect. As accurately reviewed by Taranejoo et al. [53], any increase in molecular weight or chemical stability, produce increments PEI toxicity; being only PEI chains below 1 KDa safe for in vivo applications. For this reason, other research groups have tried to develop silica coatings based on more biocompatible polymers. For instance, Nhavene et al. evaluated two biodegradable polymers as MSNs coatings [54]. The chosen highly biocompatible polymers polycaprolactone (PCL) and chitosan (CS) were grafted onto MCM-41 MSNs upon reaction of silica with an epoxide-containing (3-Glycidoxypropyl)trimethoxysilane. As a result, both nanosystems showed great biocompatibility on MCF-7 cell line (up to 100 μg·mL^−1^) and effective transfection of α-tubulin and laminB1 siRNAs. Unfortunately, no further analyses were given on the efficacy of such nanodevices, especially when employing the non-charged PCL polymer.

Despite the advances in multilayered nanosystems, this approach does not offer protection against nucleotide degradation; being pore hosting the most convenient strategy to preserve nucleic acid integrity. The development of RNA carriers based on large-pore MSNs has been successfully implemented by decorating the internal surface of mesopores with cationic groups are able to promote nucleotide threading into mesopores. However, despite electrostatically driven loading, the pH-driven release does not fully ensure biopolymer delivery on target. For this reason, Rosenholm and coworkers developed a promising release strategy based on intracellular glutathione-triggered (GSH) release. These authors modified the internal surface of mesopores with amino groups and cysteamine moieties which provided both the required positive charge and a disulfide bridge suitable for reductive cleavage [55]. The system was able to load antiGFP siRNA in the mesopores and produce effective gene silencing on genetically modified green-fluorescent MDA-MB-231 cells. This system was further coated with a PEI layer to favor carrier uptake, although without surpassing acceptable hemolytic levels.

On understanding the threading phenomenon, Rankin, Knutson and coworkers studied the behavior of amino-functionalized MSNs as carriers of double-stranded (ds) RNA oligonucleotides as a function of size (84 base pairs dsRNA with a 2.6 nm × 24 nm estimated size vs. 282 base pairs dsRNA with a 2.6 nm × 80 nm estimated size) and pore diameter (nonporous, 2.7, 4.3, and 8.1 nm pored silica particles) [56]. According to their results, both dsRNA were able to thread in all mesopores except in the case of 282 bp dsRNA which could not be loaded in the 2.7 nm wide mesopores, which set the basis for the further development of nucleotide nanocarriers able to circumvent the effect of nucleases. Nevertheless, in our opinion, this study must be completed with PEG modified MSNs more suitable for in vivo applications.

Another promising strategy for gene delivery is remote triggering. This topic was studied by Du et al., who designed an ultrasound-triggered device for plasmid DNA (pDNA) delivery [57]. Their system was built from pDNA encapsulated within mesopores (ca. 3.6 nm) of PEI-modified magnetic MSNs [58] which were further embedded within a solid lipidic microbubble able to be exploded in the presence of ultrasound activation. The resulting non-viral vector presented good biocompatibility due to the outer lipidic coating, good DNA binding stability due to the double protective layer and a remarkable magnetic targeting response. Nevertheless, despite the active magnetic targeting and the focused activation of microbubbles, this strategy must be still studied as no information is available on bioaccumulation and trafficking in vivo, especially when its size does not suggest intravenous applications.

**Table 1 pharmaceutics-15-00351-t001:** Reported devices based on the combination of silica nanoparticles and nucleotides.

Nanosystem	Assembly Strategy	Nucleotide	Secondary Therapeutic	Therapy	Release Stimulus	Action Mechanism	Biological Evaluation	Ref.
Strategies for carrying oligonucleotides and plasmids with silica nanosystems								
LP-MSNs	Electrostatic deposition (PEI)	Plasmid (eGFP)	None	None (GFP transfection)	pH-driven detachment	Transfection	In vitro: HEK-293	[50]
US-MSNs	PEI grafting (glutaraldehyde)	siRNA (siGADPH)	None	None (Knockdown quantification)	pH-driven detachment	Gene silencing	In vitro: HeLa, HEK-293	[51]
Asymmetric MSNs (Phosphonate coated)	Electrostatic deposition (PEI)	Plasmid (eGFP)	None	None (GFP transfection)	pH-driven detachment	Transfection	In vitro: HEK-293T	[52]
MSNs (MCM-41)	Chemical grafting (PCL or CS)	siRNAs (α-tubulin, laminB1)	None	None (cytoskeleton reduction)	pH-driven detachment	Gene silencing	In vitro: HeLa, MCF-7	[54]
LP-MSNs	Pore loading (Cystamine-PEI coated)	siRNA (anti-GFP)	None	None (Green florescence knockdown)	Redox (GSH) driven cleavage	Gene silencing	In vitro: MDA-MB-231	[55]
MSN@ lipid Microbubbles	Pore loading	Plasmid (eGFP)	None	None (GFP transfection)	Ultrasound	Transfection	In vitro: SKOV3, HEK-293T In vivo: Mice	[57]
Strategies for gene therapy employing silica nanoparticles as non-viral vectors.								
MSNs	Pore loading	siRNA (anti-miR-155)	None	AS1411-targeted oncogene silencing	pH-sensitive PDA coating.	Adjuvant gene silencing for 5-FU chemotherapy	In vitro: SW480, HT-29, SW620, Lovo, Caco-2, NCM460 In vivo: Mice	[59]
MSNs	Electrostatic deposition (PEI)	siRNA (anti-HER2)	None	Transtuzumab-targeted oncogene silencing	pH-driven siRNA detachment	Gene silencing	In vitro: BT474	[60]
MSNs	Pore loading	siRNA (anti-MDR1)	None	TAT-targeted anticancer siRNA delivery	Detachment of chitosan protecting layer	Gene silencing	In vitro: HeLa, EPG85.257	[61]
CC-MSNs	Electrostatic deposition (PEI)	miRNA (rno-miRNA-26a-5p)	None	Osteogenic therapy	None	Gene expression	In vitro: rBMSC	[62]
Dual MSNs (Amino modified)	Pore loading	Plasmids (GLP-1AR and FGF-21)	None	Antidiabetic therapy	None	Transfection	In vitro: Hepa1-6 In vivo: Mice	[63]
Rambutan-like MSNs	Electrostatic deposition (PEI)	Plasmid (OVA)	None	Vaccine	pH-driven detachment	Transfection of an immune stimulator	In vitro: HEK-293 In vivo: Mice	[64]
Porous silica microrods	Electrostatic deposition (PEI)	Plasmid (OVA)	None	Vaccine	pH-driven detachment	Transfection of an immune stimulator	In vivo: Mice.	[65]
Ca-dopped Si NPs (amino modified)	Not determined	Plasmid (PRPF31-GFP)	None	Blindness treatment	pH-driven detachment	Transfection	In vivo: Mice	[66]
Dendritic MSNs (amino modified)	Not determined	Antisense plasmid (ASvicR)	None	Antibiofilm (anticaries)	pH-driven detachment	Transfection	In vitro: S. mutans UA159	[67]
Strategies for combined gene and chemotherapy: siRNA delivery								
MSNs (amino modified)	Electrostatic deposition	siRNA (siVEGF)	Sorafenib	LA-targeted siRNA-drug combination	pH-driven detachment	Gene silencing plus kinase inhibition	In vitro: Huh7, HepG2, HeLa and A549	[68]
MSNs	Electrostatic deposition (PEI)	siRNA (Survivin)	Docetaxel, etoposide and/or carfilzomib	siRNA-drug combination	pH-driven detachment	Gene silencing plus proteasome inhibition	In vitro: HEK-293, A549	[69]
Disulfide bridged MSNs	Electrostatic deposition (HA-PEI)	siRNA (Bcl-2)	Doxorubicin	siRNA-drug combination	pH-driven detachment, GSH-driven SiO2 degradation	Gene silencing plus chemotherapeutic	In vitro: MCF7	[70]
Ca2+ doped LP-MSNs	Pore loading (Ca2+)	siRNA (Bcl-2)	Chloroquine	siRNA-drug combination	Pore release	Gene silencing plus chemotherapeutic	In vitro: SKOV3	[71]
Strategies for combined gene and chemotherapy: plasmid delivery								
UCNP@MSNs (amino modified)	Electrostatic deposition (H2A)	Plasmid (p53)	Bortezomib	H2A-targeted, combined genic and chemotherapy	pH-driven detachment	Transfection plus chemotherapeutic	In vitro: NCI-H1299 and HeLa	[72]
MSNs	Electrostatic deposition onto pH-labile polymer coating	Plasmid (p53)	5-Fluoruracil	Combined genic and chemotherapy	pH-driven detachment	Transfection plus chemotherapeutic	In vitro: MCF-7	[73]
MSNs	Electrostatic deposition (PEI)	Plasmid (HNF4α)	Cisplatin	Combined genic and chemotherapy	pH-driven detachment	Transfection plus chemotherapeutic	In vitro: Huh7 In vivo: Mice	[74]
Rod-shaped MSNs	Not Specified	Plasmid (Survivin)	Camptothecin	AS1411-targeted, Combined genic and chemotherapy	Not Specified	Transfection plus chemotherapeutic	In vitro: C26, CHO In vivo: Mice	[75]
Strategies for combined gene and chemotherapy: miRNA delivery								
MSNs	Lipid coating (erythrocytes’ membranes)	miRNA (miR137)	Indocyanine green (PDT)	RGD-targeted photothermal- miRNA	Detachment of protecting lipid layer	Photothermal and gene expression therapy.	In vitro: U87, RAW264.7 In vivo: Mice	[76]
MSNs	Pore loading	miRNA (hsa-miR-200c) siRNA (anti-Plk1)	Indocyanine green (PDT)	RGD-targeted photothermal- miRNA-siRNA	Detachment of protecting lipid layer	Photodynamic, gene silencing and expression	In vitro: MDA-MB-231 In vivo: Mice	[77]
MSNs	Pore loading	Dual miRNA (miR-34a and anti-miR-10b)	None	HA-targeted dual miRNA-combination	Detachment of protecting polymer layer (HA-PEG-PLGA)	Dual gene modulation	In vitro: MDA-MB-231, MDA-MB-468, HEK-293T, 4T1 In vivo: Chicken embryo In vivo: Mice	[78]
Strategies employing silica nanoparticles for gene therapy in bone diseases								
MSNs	Electrostatic deposition (PEI-PLL)	Plasmid (BMP-2)	Dexamethasone	Targeted and combined anti- osteoporotic treatment	pH-driven detachment	RGD-targeted osteoporosis gene silencing plus osteogenic stimulation	In vitro: HEK293T, BMSCs and RAW 264.7	[79]
MSNs	Pore loading	Anti-miRNA (miR26)	None	Targeted anti-osteoporotic treatment	pH-driven detachment	KALA-targeted osteogenic stimulation	In vitro: rBMSCs	[80]
LP-MSNs	Pore loading	Anti-miRNA (miR26)	None	Targeted anti-osteoporotic treatment	Pore release	Osteogenic stimulation	In vitro: rBMSCs	[62]
MSNs	Electrostatic deposition (PEI)	siRNA (SOST)	Osteostatin	Combined anti- osteoporotic treatment	pH-driven detachment	Osteoporosis gene silencing plus osteogenic stimulation	In vitro: MEF In vivo: Mice	[81]
MSNs	Electrostatic deposition (PEI)	siRNA (SOST)	Osteostatin	Combined anti- osteoporotic treatment	pH-driven detachment	Alendronate-targeted Osteoporosis gene silencing plus osteogenic stimulation	In vitro: MC3T3-E1 In vivo: Mice	[82]
Other therapies								
MSNs	Electrostatic deposition	Plasmid (RhoG)	Curcumin	TAT-targeted neurite growth	pH-driven detachment	Neurite growth induction plus antioxidative effect	In vitro: N2a cells	[83]
MSNs	Electrostatic deposition	Anti-miRNA (miR33)	None	Metabolic lipid disorder treatment	pH-driven detachment	miRNA scavenging	In vitro: L02, LX02, RAW264.7 In vivo: Mice	[84]
LP-MSNs	Electrostatic driven loading	Plasmid (antisense vicR)	None	Biofilm disruption	pH-driven detachment	Reduction of extracellular polysaccharides	In vitro: *Streptococcus Mutans*	[67]

Abbreviations: BMDCs: Bone Marrow Dendritic Cells; CC-MSNs: Core-cone Mesoporous Silica Nanoparticles; CS: Chitosan; GFP: Green fluorescence protein; GSH: Glutathione; H2A: Histone 2A; HA: Hyaluronic acid; LA: lactobionic acid; LP-MSNs: large pore mesoporous silica nanoparticles; OVA: Chicken Ovalbumin; PCL: Poly-ε-caprolactone; PDA: polydopamine; PEI: polyethylene imine; PLL: polylysine; UCNPs: Upconversion nanoparticles; US-MSN: Ultrasmall mesoporous silica nanoparticles; rBMSC: rat bone marrow mesenchymal stem cells.

### 2.1. Gene Therapy and Gene Silencing

One of the most active areas of non-viral vectors research is gene therapy and silencing for cancer treatment, but not the only one as discussed below. The relevance of gene therapies for anticancer treatment relies on the genetic origin of tumors, which can be modulated with siRNAs and miRNAs to reduce disease’s virulency and improve survival. In this section, we will focus on the most recent advances obtained for silica-based gene therapies employing oligonucleotides. However, as the porous structure of silica enables the loading of additional chemotherapeutic agents, it is logical to assume that most promising anticancer strategies result from drug-loaded and multifunctional nanosystems. Combination therapy, due to its importance and number of contributions during the last years, will be reviewed in a following section. In fact, pioneering works on MSNs as non-viral transfection and gene silencing were developed two decades ago [38] and have been visited in numerous contributions. For a review focused on the insights of such combinations, we recommend the reader a previous contribution by us in which many details were covered [39].

Nevertheless, during the last 5 years, significant advances also have been made on nanosilica-based gene therapies (Table 1). Regarding anticancer treatments, there have been reported interesting advances on targeted nanodevices as well as in combination therapies. Along these lines, Li, Duo and coworkers, reported an aptamer-targeted nanosystem for oncogene silencing therapy [59]. In this work the authors focused their strategy in silencing the oncogenic microRNA miR-155, known to upregulate many human cancers. The loading of the therapeutic anti-miR-155 siRNA (5′ACCCCUAUCACGAUUAGCAUUAA3′) was performed onto amino-modified MSNs (ca. 125 nm) with regular mesopores (2.8 nm), which were further coated with a PDA layer suitable to graft the AS1411 nucleolin-targeted aptamer. The resulting system was able to effectively repress the expression of miR-155 and NF-κB in SW480 cells and accumulate within the tumor in vivo after 72 h (during the first 48 h it was mainly accumulated in the lungs). This treatment, especially in combination with 5-fluoruracil (5-FU) made it possible to slow down tumor growth, although did not achieve a complete growth arrest. A gene silencing effect was demonstrated upon an effective downregulation of the PGP protein which sensitized treated mice against 5-FU.

Another interesting targeted nanosystem aimed at anticancer gene silencing was reported by Ngamcherdtrakul, Sangvanich and coworkers [60]. These authors developed a nanoconjugate based on MSNs (50 nm) coated with cross-linked PEI, a PEG spacer and at the outermost layer the trastuzumab monoclonal antibody (MAB). The resulting MSN@PEI@PEG-MAB nanodevice was suitable to bind siRNAs throughout a multilayered assembly. The system was first evaluated against luciferase activity using a siLUC siRNA (5′ CGGAUUACCAGGGAUUUC-Att 3′) and then with siHER2 (5′ CACGUUUGAGUCCAUGCCCAAUU 3′) RNAs against the BT474 cell line. The system achieved a reduction in cell viability close to ca. 80%, which set an interesting starting point for the development of more advanced therapies. Beyond the interest provided by the potential of this nanosystem, it is also remarkable the freeze–thaw processing of MSN@PEI particles; as typically the combination of PEI and silica usually leads to high degradation rates in aqueous media. Herein, the use of a cytoprotectant makes it possible to maintain their structural integrity of MSN@PEI@PEG-MAB for long periods of time and their further combination with siRNA on demand.

To conclude with examples dealing with anticancer siRNA delivery, i is worth citing the work by Heidari et al., which proposed a chitosan-coated, dual-targeted nanosystem to reduce the anticancer multidrug-resistant pump MDR1 [61]. This system employed amino-modified MSNs (70 nm with 5 nm mesopores) were loaded with the anti-MDR1 siRNA and a later a chitosan coating layer responsible for retaining the cargo serve as anchoring point for targeting groups. The functionalization of such chitosan layer with a Maleimide-PEG3000-NHS bifunctional linker made it possible to covalently bind NH_2_ groups of chitosan to thiol groups from cysteine groups at TAT peptides to thiol modified folic acid moieties. The overall performance of this nanosystem was evaluated against HeLa and EPG85.257 cells with MDR1 overexpression. As a result, a significant reduction in MDR1 levels was achieved without affecting viability, which suggests that this kind of nanosystem could be employed as chemotherapeutic adjuvants by reducing the chemotherapeutic outflux from cancerous cells.

Beyond the use of siRNA delivery against cancer, there are also interesting approaches for the treatment of other genetic diseases. For instance, Hosseinpur et al. employed miRNA delivery to enhance osteogenicity. These authors employed large pore sized (40 nm) core-cone silica nanoparticles (ca. 200 nm) to load the rattus norvegicus (rno) rno-miRNA-26a-5p with the aid of PEI [62]. The system proved to have acceptable biocompatibility before miRNA loading and once loaded proved to boost the expression of seven genes associated with osteogenesis (Runx-2, OCN, Col1α1, IBSP, GSK3β, ALPL, and BMP-2) together with the production of collagen and alkaline phosphatase (ALP) which are associated with bone formation.

Another interesting example, reported by Niu, Yang and coworkers, focused on the treatment of diabetes. These authors created a nanosystem able to transfect two plasmids: the fibroblast growth factor 21 (FGF-21, known to improve glucose metabolism and insulin resistance) and Liraglutide, an analog of the Glucagon-like peptide-1 (GLP-1) with a hypoglycemic effect [63]. The nanosystem, based on hierarchically porous silica nanospheres [85], was able to load and deliver both therapeutics, and in particular the FGF-21-containing plasmid with higher efficacy than Lipofectamine 2000. In vitro, treated mice improved glucose tolerance, lowered blood glucose levels and reduce body weight without significant toxicity or side effects; which suggest a potent long-lasting effect caused by the effective delivery of both plasmids to cells.

Apart from the cited diseases, during the last years cancer immunomodulation has also become a very active field of research. Many of the reported examples employ an immunostimulant protein, typically chicken ovalbumin (OVA), to trigger an immune response against such xenoprotein. The direct consequence is the generation of an immune response against the tumor-forming cells and thus achieve anticancer vaccination. For an in-depth discussion on the topic, please refer to the following section and to reference [86]. Unfortunately, the continuous delivery of xenoproteins intratumorally is not an easy task, for which reason research groups have explored another possibility: plasmid transfection. Ideally, this will produce a sustained immune response at the tumoral region. Along this line two recent examples by Nguyen et al. and Song et al. reported the use of mesoporous silica for the delivery of OVA encoding plasmid [64,65]. In the first example, the authors employed Rambutan-like MSNs [64], a particular kind of particle with a unique spiky surface that, as highlighted above, shows superior nucleotide delivery efficiency, whereas, in the second example the authors employed a branched PEI (60 KDa) to bind the OVA pDNA into a polyplex that was latter supported on rod-shaped mesoporous silica microparticles [65]. With obvious differences on delivery kinetics consequence of particle sizes, both nanosystems were demonstrated to induce effective immune activation as demonstrated by the overexpression of typical markers: CD86, TNF-α, IFN-γ, IL-12, and IL-4. In vivo assays on mice demonstrated that plasmid-loaded, Rambutan-like MSNs, accumulate within the tumor, which in the absence of further results, suggests the possibility of achieving effective antitumor vaccination.

In addition to previous examples, in which nanosystems are designed for intravenous or intraperitoneal dosage, there are also interesting aimed at local dosage. As a representative examples of this strategy, there could be highlighted the nanodevice prepared by Valdés-Sánchez et al. to prevent blindness caused by retinitis pigmentosa [66]. In this work, the authors prepared amino-modified, Ca-dopped silica nanoparticles (ca. 150 nm with ≈3.5 nm mesopores) employing a sol–gel method onto which loaded the PRPF31-GFP plasmid. Upon in vitro evaluation, the system was evaluated in vivo throughout subretinal dosage in mice. Treated animals maintained the integrity of the retinal tissue and the visual function, which suggests an effective gene delivery by the silica-based non-viral vector and thus a protective effect against retinal degenerative diseases.

Another interesting example of local administration was reported by Tian et al., who developed a dendritic silica nanosystem able to deliver an anti-*Streptococcus mutans* plasmid [67]. Employed particles were prepared with cyclohexane as pore expander and an amino-modified surface to ensure electrostatically binding of the plasmid. The integrity of the DNA was ensured by treating the system against DNase I, proving that the multilayered structure gave some protection against degradation. When tested against S. Mutans, the nanosystem was able to deliver the antisense vicR plasmid, disrupting the normal function of bacterial enzymes responsible for wall biosynthesis. Consequently, although this local treatment did not result in destruction of bacteria, it made it possible to significantly disrupt the synthesis and function of biofilm, thus sensitizing these cariogenic bacteria against conventional antibiotics.

### 2.2. Combined Anticancer Therapies

Despite advances made in clinical practice, to date, chemotherapy remains the main therapeutic tool for the treatment of cancer. The well-known side effects occurring as a consequence of its use, especially the appearance of multidrug resistance (MDR) in relapsing cancers, demand more selective and active treatments. Combination therapy seems to improve overall efficacy through synergistic effects, although at the expense of maintaining chemotherapy’s typical side-effects. Apart from the classical combination of chemotherapy with surgery or radiotherapies, during the last years new promising nanotherapies have also emerged. Among them, (1) immunotherapy promises to recruit the patient’s immune system to destroy cancer cells, while (2) gene therapy claims to either overcome drug resistance by disrupting active immortality pathways of cancerous cells. In fact, both strategies have been successfully employed in the clinic and in particular the combination of chemotherapeutic drugs and siRNAs has proven to be a valuable combination against cancer [87].

However, profiting from this combined anticancer effect is not easy, as the stability of siRNA-based therapeutics is very short unless efficiently delivered. In addition, secondly, the differences on pharmacokinetics of both chemotherapeutic drug and siRNA makes it extraordinarily difficult to achieve a simultaneous action. For these reasons, nanoparticles have arisen as promising platforms to deliver both agents simultaneously and overcome one of the limitations of combination therapy. Since our first review on this topic [88], the number of formulations has grown significantly. In fact, the current state of the art is split between (1) the clinician’s point of view, focused on ultra-specialized applications of particular drug-siRNA combinations, as can be guessed from reviews with titles such as Overcoming doxorubicin resistance in cancer: siRNA-loaded nanoarchitectures for cancer gene therapy [89] and Employing siRNA tool and its delivery platforms in suppressing cisplatin resistance: Approaching to a new era of cancer chemotherapy [90]; and (2) the materials scientist’s point of view, which is more focused on the development of strategies and understanding mechanisms of action of such nanosystems, like the excellent review by Paris and Vallet-Regí [91].

Among the strategies reported, multiple chemotherapeutic agents, oligonucleotide sequences, and combinations of both can also be found. For instance, Zheng et al. reported the use of MSNs to deliver Sorafenib, a kinase inhibitor in combination with a Cy5-labeled Vascular Endothelial Growth Factor (VEGF) siRNA (sense: 5’-GGAGUACCCUGAUGAGAUCdTdT-3’) [68]. Their system was built employing Sorafenib-loaded, amino-capped MSNs onto which the siRNA was electrostatically deposited and the Lactobionic acid (4-O-β-D-galactopyranosyl-D-gluconic acid, LA) grafted. The resulting nanosystem showed good antiproliferative effect on four typical cancer cell lines (Huh7, HepG2, HeLa and A549), showing a synergistic behavior when both anticancer compounds were combined. In the best cases, the authors reached more than 80% reduction on viability at 60 μM concentration of particles. Another interesting example was reported by Dilnawaz and Sahoo [69]. These authors evaluated the behavior of single-drugs vs. drug combinations, both delivered by MSNs or in their free form, against healthy HEK-293 and lung adenocarcinoma A549 cell lines. According to their results, MSN-loaded combinations of Docetaxel or Etoposide with Carfilzomib showed the best effect. In a second step, they evolved their nanosystem by including a Survivin siRNA (Sense: 5’-GGACCACCGCAUCUCUACAdTdT-3’) throughout a sequential PEI-oligonucleotide coating. The evaluation of the resulting combined drug nanosystem made it possible to reach cell apoptosis up to ca. 50% due to the simultaneous knockout of Survivin and dual action of drugs. Unfortunately, no further evaluation was done in vivo to fully prove their suitability as therapeutic models.

Apart from the previous examples, interesting efforts have been also made on improving carrier biocompatibilities and release rates. For instance, Zhang et al. reported the use of a biodegradable nanosystem based on reducible coating and silica matrix to co-deliver Doxorubicin (DOX) together with an anti Bcl-2 siRNA (sense 5’-CGGGAGAUAGUGAUGAAGdTdT-3’) [70]. Prepared nanoparticles contained disulfide bridges along their structure, which are quickly cleaved in intracellular environments. The advantages of these particles go beyond fast degradation, as they show also quick cargo delivery and may induce deregulation of homeostasis throughout depletion of the cytoprotective Glutathione, which may also have an additional adjuvant effect. To build the system, the authors employed a layer-by-layer assembly strategy using hyaluronic acid as the first coating layer and PEI as a cationic compound. Data shown suggested that the nanosystem designed for DOX/Bcl-2 siRNA co-delivery led to higher rates of late apoptosis in MCF-7 cancer cell line (>99%) than those yielded with carried DOX (59%) or siBcl-2 (10%) alone, which suggest an effective gene/chemo-synergic therapy. Another interesting example was reported by Choi et al. To improve biocompatibility these authors employed Ca2+ doped MSNs to create an amino-free nanosystem able to co-deliver the anti Bcl-2 siRNA (sense: 5’-GUACAUCCAUUAUAAGCUGdTdT-3’) directly bound to MSNs’ surfaces due to the presence of Ca cations and Chloroquine as chemotherapeutic [71]. The resulting system showed a sustained but pH-dependent release of siRNA and an effective downregulation of Bcl-2 expression on SKOV3 cell lines, yielding similar values as those reported with the lipofectamine reference. The presence of Chloroquine in the nanosystem also helped in reducing cell viability, reaching up to 30% of reduction, which although not outstanding from a therapeutic point of view, confirms the efficacy of this nanoformulation.

In addition to gene silencing, several research groups have also explored combined therapies which typical chemotherapeutics and plasmids, being the p53 tumor suppressor protein and Survivin, inhibitor of caspases’ pathways, two of the most promising candidates for reverting tumoral profile. Along this line, Rong et al. and Zhou et al. reported the combination of a p53 plasmid with typical chemotherapeutic drugs to obtain double action nanotherapeutics. In the system by Zhou et al., the authors managed to load the hydrophilic drug 5-fluoruracil within the mesopores of silica nanoparticles [73]. Due to the difficulty of this process, the chemotherapeutic drug was entrapped within the mesopores with the aid of a cationic, acid-cleavable block copolymer based on 2-(dimethylamino)ethyl methacrylate (DMAEMA). The resulting nanosystem proved to have an outstanding biocompatibility with a slight reduction of viability up to 200 μg·mL^−1^ in MCF-7 cell line and hemolysis rates below 5%. This model only yielded a modest cell viability reduction (about ca. 50%) despite the dual effect and the success of p53 plasmid transfection. On the other hand, the nanosystem reported by Rong et al. employed Bortezomib (BTZ) as cargo. Chosen core–shell upconversion-silica nanohybrids were modified through a grafting step with Histone 2A as targeting moiety and finally coated with the p53 plasmid electrostatically [72]. In this case, the protein, apart from favoring migration to nucleus [92] showed additional features: (1) increased biocompatibility (augmented cell viability of empty carrier); (2) reduced hemolysis rate; and (3) enabling of the subsequent interaction with the p53 plasmid. The complete system was able to efficiently transfect p53 into cells’ nucleus, and in combination with BTZ, made it possible to reduce HeLa and NCI-H1299 cells viabilities about ca. 65% and 80% respectively, demonstrating that nuclei targeting must be considered too for the development of future nanoformulations.

In addition to previous examples, there are other combinations of plasmids and chemotherapeutic drugs reported in the literature. For instance, Tsai, Wang and coworkers [74] reported a system based on the combination of Survivin plasmid with Camptothecin, while Alibolandi and coworkers tested a combination of HNF4α plasmid and Cisplatin [75]. In the first example, amino-modified, rod-shaped MSNs were employed to create targeted nanoparticles upon grafting of the AS1411 aptamer. Then, onto the surface, the remaining unfunctionalized amino made it possible to undergo electrostatic deposition of the negatively charged plasmid to provide a transfecting system with nuclear-targeting properties. The combination of both substances showed a clear synergistic effect against C26 cancerous cells (ca. 80% viability reduction), while healthy CHO cells were not affected in this manner (ca. 30% viability reduction). Further in vivo experiments also demonstrated that the targeting agent was crucial to obtain an adequate intratumor distribution, while in its absence bioaccumulation occurred preferentially in liver and kidney. In the second example, the absence of any targeting element was justified because the final purpose of such nanosystem was a liver disease. In this model, negatively charged phosphonate-capped cisplatin-loaded MSNs were coated with a 2 KDa PEI and the corresponding plasmid. This system was able to reduce the tumorigenic capacity of Huh7 cells according to data shown, achieving even effective tumor volume reduction when both therapeutic agents were combined in the nanodevice.

In addition to plasmid transfection, the use of RNAs also offers broad potential for the development of temporary therapies. The use of micro RNAs has been reported alone and in combination with other cytotoxic compounds. For instance, Li et al. reported the use of lipid-coated MSNs to deliver the miR-137 RNA (sense: 5’-UUAUUGCUUAAGAAUACGCGUAG-3’) in vitro and in vivo [76]. Herein, to improve biocompatibility and efficacy, the authors employed three complementary strategies on their design: (1) the external coating was fabricated from red blood cells membranes’, which were (2) modified with RDG targeting moieties to enable targeting and (3) included the indocyanine green photosensitizer within the porous silica structure. As a result, the system was able to reach adequate intratumor accumulation due to the RDG targeting and the advanced immune cloaking. In vivo experiments showed that combined photothermal and gene therapy made it possible to treat solid tumors so efficiently that even tumor reduction was observed. A similar design employing RGD-targeted, lipid-coated MSNs was also reported by Oupický and coworkers, who employed a silica-based nanodevice to deliver the siPlk-1 anticancer RNA (sense: 5′-CAACCAAAGUCGAAUAUGAUU-3′) in combination with a microRNA-200c mimic (sense: 5′-UAAUACUGCCGGGUAAUGAUGGA3′) [77]. Their system, which also contained indocyanine green as a photosensitizer, was also able to destroy tumors in vivo, achieving complete tumor remission in mice and an antimetastatic activity in an orthotopic breast cancer model highlighting the potential of gene therapy in combination therapies.

As discussed above, the combination of therapeutic RNAs makes it possible to develop highly interesting combined therapies; although for such purposes the use of chimeric RNAs [93] would be of interest as combining several effects in a single nucleotide strand would facilitate delivery processes. However, the combination of different RNA sequences may be of interest too if different dosages are required. Along this line, Ahir et al. explored the potential therapeutic effect of the hsa-miR-34a-5p miRNA (sense: 5’- UGGCAGUGUCUUAGCUGGUUGU-3’) together with the anti-sequence miR-10b-5p (5’-CACAAAUUCGGUUCUACAGGGUA-3’) [78]. This system [78] was prepared from trimethylammonium-modified MSNs, which were able to coordinate both RNA strands via electrostatic interactions. Then, to conclude the system was coated with a pore-blocking PLGA layer and hyaluronic acid sequences with targeting capabilities. The complete nanosystem was able to interact with the CD44 receptor, internalize into cancer cells and promote an early apoptotic state. The external polymeric layer made it possible to accomplish in vivo studies in both mice and chicken embryo with satisfactory results, due to the great biocompatibility provided. However, upon tumor delivery, cell uptake and system disassembly, both therapeutic RNAs made it possible to achieve significant tumor reduction, although in this case, the polycationic nanoparticle may have membrane-lytic effect too and may have a contribution on the overall antitumor effect.

### 2.3. Gene-Based Therapies against Bone Diseases

As outlined previously, gene editing allows the treatment of many diseases of genetic origin in addition to cancer; as reviewed by Knežević and coworkers [94]. Among such diseases, bone ones have recently attracted great attention; especially those in combination with ceramic materials such as silica, which offer great osteocompatibility and interesting properties as filling material. On this topic, He and coworkers explored the antiosteoporosis effect of a plasmid encoding the bone morphogenetic protein 2 (BMP-2) (pIRES2-ZsGreen1-BMP2) in combination with the dexamethasone glucocorticoid [79]. This nanodevice employed PEI-PLL as a connecting layer due to the better biocompatibility and higher transfection efficacy than pure 25 KDa PEI. Moreover, to enable cell targeting, the outermost surface of the resulting carrier was modified with the RGD motif and thus promote cell internalization. Drug and plasmid were integrated within the nanosystem throughout sequential pore loading and DNA surface electrostatic deposition onto the positively charged polymer-coated particle. Cell cultures on Bone Marrow Mesenchymal Stem Cells (BMSCs) showed effective transfection and drug release, which yielded an overexpression of the BMP-2 protein along with other typical bone formation markers such as Alkaline phosphatase (ALP) and the osteo-related genes RUNX2, OPN, Col 1 and OCN in vitro, demonstrating the effective action of both components.

The use of miRNAs also has positive effects, as reported by Yan et al. [80] and Hosseinpour et al. [62]. These authors explored the efficacy of silica-carried 26A miRNA in the osteogenic differentiation of rat BMSCs, although using complementary delivery strategies. In the first example, the RNA was deposited onto positively charged MSNs@PEI, while in the second, the RNA was loaded within the porous structure of MSNs. As a result, both nanosystems were able to deliver the miRNA intracellularly, although the first model, modified with the cell-penetrating KALA peptide, showed better performance and improved biocompatibility. However, both nanoformulations were able to easily achieve 3-fold increases in overexpression of osteo-related genes; although the in-pore-loaded miRNA nanosystem showed higher BMP-2 expression (ca. 10-fold vs. 3.5-fold) due to the enhanced protection offered by the silica matrix in these for single-drug biopolymer delivery.

In addition to previous examples, Manzano, Vallet-Regí and coworkers also explored an osteoporosis treatment based on targeted nanotherapeutic able to deliver simultaneously an osteogenic peptide and an anti-osteoporotic siRNA. In their first formulation, the authors evaluated the codelivery of siRNA and peptide combination [81], while in a later contribution, they incorporated the alendronate bone-specific targeting moiety to improve efficacy [82]. Both formulations were prepared employing MCM-41-like MSNs, which upon peptide loading were reacted with PEI and the anti-SOST siRNA to obtain the final device. The effective silencing of SOST expression made it possible to obtain upregulation of both ALP (Alkaline phosphatase) and RunX2 proteins, responsible for bone regeneration both in vitro and in vivo. Unfortunately, their first model was only effective upon highly invasive bone injection; although in the evolved model, an alendronate-targeted nanosystem achieving satisfactory antiosteoporosis effect in female ovariectomized mice could also be achieved, but with subcutaneous dosage. In this nanoformulation, the bisphosphonate-containing alendronate was linked to the silica matrix through a PEG connector, also allowing the loading of osteostatin and siRNA, as previously described.

### 2.4. Other Gene-Based Therapies Employing Silica Particles as Nucleotide Carriers

In addition to the discussed examples, nucleotide-based therapy using silica-based nanocarriers could be also employed for the treatment of other genetic diseases. However, as consequence of the limited number of reported examples, these must be considered as proofs of concept rather than as potential therapeutics. For instance, Wu, Chen and coworkers reported a nanosystem able to prevent neuron damage caused by chronic inflammation in neurodegenerative disease. For so, the authors designed a multifunctional nanosystem composed by an antioxidant (curcumin) and a neurite growth promoter (RhoG containing plasmid) and the TAT cell-penetrating peptide (AYGRKKRRQRRR) to favor cell uptake [83]. The resulting device was able to transfect Na2 cells favored by the positively charged targeting peptide, as demonstrated by the transfection of a red fluorescent protein and induce neurite growth, and once internalized, induce cell growth throughout a dual action of the plasmid and the curcumin. In fact, both effects were determined separately employing Paraquat, a known generator of superoxide anion radicals, and the non-therapeutic GFP plasmid.

Another example of gene-originated disorders that could be treated with silica-based nanomaterials was reported by Tao et al., who designed a nanosystem able to deliver a microRNA-33 antagomir to prevent lipid metabolic disorders [84]. Herein, cholesterol-targeted amino-modified MSNs were loaded with an antagonist of the miRNA-33 (sense: 5’UGCAAUGCAACUACAAUGCAC-3’). As a result, the system could internalize into hepatic cells and promote lipid metabolism, thus reverting fatty liver condition. In vivo studies showed that treated mice had an accelerated lipid metabolism, as determined by the lower weight gain observed. In fact, body weights of treated mice showed similar values to lovastatin treated mice, pointing out an effect of similar magnitude to that obtained by conventional antilipidemic drugs.

In conclusion, plasmid delivery has also been employed to treat bacterial infections. Through a downregulation of extracellular polysaccharides in Streptococcus Mutans colonies, Tian et al. were able to reduce the cariogenic potential of this type of cocci [67]. Their model was built by electrostatic deposition of the negatively charged anti-VicR plasmid onto dendritic amino-modified MSNs. This particular structure of silica showed to be of capital importance to obtain adequate protection against DNA degradation and transfection into bacteria. Genetically disrupted cocci showed a reduced formation of extracellular matrices which led to a weaker and disordered biofilm, making them highly sensitive to conventional antibiotics and reducing their potential carcinogenicity.

## 3. Protein- and Peptide-Based Therapeutics Employing Silica-Based Nanosystems

As discussed previously, proteins have capital roles in many physiological processes. However, oppositely to nucleic acids, any possible therapeutic effect of proteins relies on how they reach their destination, as denaturalization may occur. For that reason, it is important to develop carriers that are able to keep their structural integrity during trafficking [95,96]. Because of the need for having such protective features, protein delivery has become a popular topic during the recent years. However, associated with the intrinsic complexity of such delivery, clear strategies to accomplish successful dosages are still unavailable and thus many authors still explore the possibilities of porous silica. For interested readers, the following reviews and articles deal with the insights of protein loading and delivery with MSNs [97,98,99]. On the other hand, peptides have attracted great attention too due their simplified structures that can partially retain functional features. Such biological activities in combination with a very convenient chemical robustness is of interest to produce low-cost therapeutics for regulating metabolic cycles and signaling processes.

Due to the number of physiological tasks developed by proteins and peptides [100], the following section will be structured in several epigraphs, covering the following aspects: (1) anticancer and pro-apoptotic proteins and peptides [101,102,103], (2) antibacterial proteins and peptide delivery [104,105], (3) immunostimulant nanoformulations, (4) nanocarriers for enzymes, and (5) delivery of growth factors as outlined in Figure 3 and detailed in Table 2. In this review, non-therapeutic applications of peptides will be excluded. For instance, cellular targeting [106] on both eukaryotic [107] and prokaryotic [108] cells, combined therapies using non-based silica carriers [109] and peptide-based stimuli responsive materials [47,110] will not be covered, despite their relevance.

### 3.1. Anticancer Therapies

The difficulty of handling proteins, which can easily undergo denaturalization and lose their function, makes it difficult to develop interesting examples on anticancer protein delivery. Nevertheless, the adequacy of MSNs for hosting, transporting and delivering proteins was described by Lin and coworkers decades ago. They pioneered pore-enlarged MSNs to facilitate protein hosting and thus promote uptake of proteins through impermeable membranes. These authors focused on Cytochrome C (Cyt c), a relatively small protein with pro-apoptotic effect triggered through the caspase pathway [111]. Herein, unaltered MSNs with 5.4 nm mesopores could accommodate this globular protein to produce intracellular delivery, albeit without control on protein release. More recently, Griebenow and coworkers revisited Cyt c delivery with MSNs bearing thiol-modified mesopores [112]. This evolved system had cleavable bonds able to retain Cyt c within the mesopores and trigger its release upon increased levels of glutathione such as those in cytoplasmatic environments. Similarly, Shang et al. employed Cyt c as a representative example for MSN-based delivery [113]. Herein, the authors examined the loading capacity and protein activity in relation to nanoparticle sizes, rather than evaluating the overall delivery. Their findings showed that larger diameters and flatter surfaces allowed the adsorption of more protein units and yielded higher overall activities. In another interesting contribution to the topic, Davis and coworkers optimized the connecting linker used to bind proteins to particle surfaces. Employing Cyt c as model, they systematically evaluated several custom-made linkers against their most relevant criteria that affected protein delivery [114]. Among them, surface charge (>40 mV on ζ-potential), ionic charge at acidic pH, endosomal escape properties and surface retention were studied. According to their results, 1 mol% functionalization of primary amines made it possible to accomplish all those tasks successfully, being the best basis for further surface grafting strategies.

In addition to Cyt c, a broad number of anticancer proteins were also reported as payloads in nanoparticle-based delivery [96] although when dealing with MSNs, the number of examples significantly decreased.

**Table 2 pharmaceutics-15-00351-t002:** Examples of therapeutic proteins and peptides delivered by silica-based nanocarriers. For an illustrative review on the coupling protocols employed for linking peptides onto nanocarriers, please check reference [115].

Nanosystem	Assembly Strategy	Protein or Peptide	Secondary Therapeutic	Biomolecule Release Stimulus	Action Mechanism	Biological Evaluation	Ref.
Delivery of peptides and proteins in anticancer strategies							
MSNs	Pore loading	Cytochrome C	None	Pore release	Cell apoptosis promoter	In vitro: HeLa	[111,112]
MSNs	Surface adsorption		None	Electrostatic detachment		None	[113]
MSNs	Surface grafting		None	None		In vitro: HeLa	[114]
MSNs	Surface grafting	Concanavalin A	None	None	Targeting plus upregulation of metalloproteinases	In vitro: MC3T3-E1, HOS.	[116]
MSNs	Surface grafting	BAMLET	Docetaxel	None	Cell apoptosis promoter plus chemotheraputic	In vitro: COS7, U87 MG In vivo: Zebrafish In vivo: Balb/c Mice	[117]
MSNs	Surface grafting	K_8_-Citraconate K_8_(RGD)_2_	Doxorubicin	None	Membrane disruption plus chemotherapy	In vitro: COS7, U87 MG In vivo: Mice	[118]
MSNs	Surface grafting	TPP-K-(KLAKLAK)_2_-	Topotecan	None	Mitochondrial membrane disruption plus chemotherapy	In vitro: KB	[119]
MSNs	Surface grafting	C-GRK_2_R_2_QR_3_P_2_Q-RGDS C-GKGG-D(KLAKLAK)_2_	Doxorubicin	None	Membrane disruption plus chemotherapy	In vitro: HeLa, COS7	[120]
MSNs	Surface grafting	(RGDWWW)_2_KC	Doxorubicin	None	DNA-intercalation plus chemotherapy	In vitro: COS7, U87 MG	[121]
MSNs	Surface grafting	(KLAKLAK)_2_	Doxorubicin	None	Membrane disruption plus chemotherapy	In vitro: HeLa	[122]
MSNs	Surface grafting and pore loading	ε-poly-*L*-lysine (surface)	C9h (pore)	Enzymatic degradation (Pore release)	Membrane disruption plus propapoptotic induction	In vitro: HeLa	[123]
MSNs	Pore loading	RDG-Hylin a1	None	Pore release	Targeted cytolytic peptide	In vitro: HeLa Hep2 In vivo: Mice	[124]
HMSNs	Cavity loading	Pepstatin A	None	Release from cavity	Inhibition of Aspartyl protease	In vitro: MCF-7	[125]
MSNs (SBA-16)	Pore loading	Alamandine	None	Pore release	Unknown	In vitro: 4T1, A549, HEK-293	[126]
MSNs	Pore loading	Fabatin	None	Pore release	Mitochondrial disfunction	In vitro: MDA-MB-23, MCF-10A In vivo: Mice	[127]
SPION@MSNs	Pore loading	Mellitin	None	Thermosensitive–hydrolytic cleavage	Apoptosis induction and suppression of angiogenesis (VEGF)	In vitro: PANC-1 In vivo: Mice	[128]
Antibacterial therapies							
MSNs	Adsorption	Lysozyme	None	Electrostatic detachment	Bacterial wall hydrolysis	In vitro: *E. coli* In vitro: HEK-293, LO2 In vivo: Mice	[129]
MSNs	Pore loading		None	Pore release		In vitro: *E. coli*	[130]
HMSNs	Surface adsorption		None	Electrostatic detachment		In vitro: *E. coli* In vivo: Mice	[131]
HMSNs	Cavity loading		None	Cavity release		In vitro: *E. coli*	[132]
MSNs	Surface grafting	Concanavalin A	Levofloxacin	None	Glycopeptide targeting	In vitro: *E. coli*	[133]
MSNs	Pore loading	Bactofencin A	None	Pore release	Defensin-like peptide	In vitro: *S. aureus* In vitro: HEK-293	[134]
MSNs	Pore loading	β-defensin-2	None	Pore release	Defensin-like peptide	In vitro: *Clavibacter michiganensis*	[135]
Solid SiO_2_ vs MSNs	Surface adsorption vs. pore loading	LL37	None	Surface adsorption vs. pore release	Transmembrane pore formation (α-helical shape)	In vitro: *E. coli*	[136]
RSNs	Surface deposition		None	Release from rough surface	Transmembrane pore formation (α-helical shape)	In vitro: *E. coli*	[137]
MSNs	Pore loading	NZX	None	Pore release	Antituberculotic peptide	In vitro: *M. tuberculosis* In vivo: Mice	[138]
MSNs	Surface grafting	Ovotransferrin	Gentamicin	None	Membrane lysis	In vitro: *E. coli* In vivo: Mice	[139]
MSNs (phosphonate vs. raw MCM-41)	Pore loading	Pexiganan, Indolicidin, or [I^5^,R^8^] Mastoparan (Electrostatically located at the surface)	*trans*-chalcone, curcumin, quercetin or berberine chloride	Electrostatic detachment	Sortase A inhibition (small molecules) plus antibiotic peptide. Pexiganan (cationic), Indolicidin (filamentator), Mastoparan (α-helix pore formation)	In vitro: *S. aureus*, *S. aureus (Methilin resistant)*, *E. coli*, *P. aeruginosa*	[140]
Osteogenic therapies							
MSNs	Pore loading	bFGF	None	Pore release	Growth factor (fibroblast)	In vitro: HUVEC	[141]
MSNs	Surface adsorption	BMP-2	None	Electrostatic detachment	Bone morphogenetic protein	In vitro: bMSCs In vivo: Mice	[142]
MSN@SPION	Pore loading			Pore release		In vitro: bMSCs	[143]
MSNs	Pore loading	Osteostatin	None	Pore release	Hormone-related peptide	In vitro: MC3T3-E1 In vivo: Rabbit	[144,145,146]
MSNs MS-HANs	Pore loading	OGP	None	Pore release	Osteogenic growth peptide (Unknown mechanism)	In vitro: Mesenchymal stem cells	[147]
MSNs	Surface adsorption	BMP-2 derived peptide	None	Pore release	Bone morphogenetic protein derived peptide	In vitro: BMSCs In vivo: Rat	[148]
MCaSiNs	Pore loading	GL13K	Sr^+2^ doped matrix	Pore release (peptide)	Antibiotic peptide Osteoclastogenetic promoter	In vitro: HBMSCs In vitro: *S. aureus*	[149]
Immunotherapy							
HMSNs	Cavity loading	IgG	None	Cavity release	Proof of concept	In vitro: HeLa	[150]
RSNs	Interparticle loading	Cyt c, IgG, Anti-pAkt	None	Cavity release	Proof of concept	None	[151]
HMSNs	Pore loading	OVA	None	Pore release	Xenoprotein induced immunostimulation	In vitro: NIH3T3 In vivo: Mice	[152,153]
DMOHS	Pore loading		None	Pore release	Xenoprotein induced immunostimulation	In vivo: Mice	[154]
MSNs	Pore loading		CpG	Pore release	Xenoprotein induced immunostimulation plus a Toll-like Receptor agonist (CpG)	In vitro: RAW264.7 In vivo: Mice	[155]
HMSNs	Cavity-pore loading	IL2	Retinoic acid Doxorubicin	Lipid layer detachment. Multiple release	Immunostimulant protein (IL2) Chemotherapeutic (DOX) Apoptotic promoter (RA)	In vitro: L929 In vivo: Mice	[156]
MSNs	Pore loading	IL13	None	Pore release	Immunostimulant protein (IL13)	In vitro: BMDMs In vivo: Mice	[157]
HMSNs	Surface adsorption	ORF2	None	Electrostatic detachment	Anti circovirus vaccine	In vitro: PK15 In vivo: Mice	[158]
MSNs	Surface adsorption	SWAP	None	Electrostatic detachment	Anti-parasite vaccine	In vivo: Mice	[159]
MSNs	Surface adsorption	HSP70	None	Electrostatic detachment	Anti-mycoplasma vaccine	In vivo: Mice	[160]
MSNs	Surface adsorption	EspA	None	Electrostatic detachment	Anti *E. coli* vaccine	In vivo: Mice	[161]
MSNs	Surface adsorption	rPb27	None	Electrostatic detachment	Anti-fungi vaccine	In vitro: HEK-293 In vivo: Mice	[162]
HMSNs	Cavity loading	TRP2 (cavity)	HGP100 (pores)	Lipid layer detachment. Dual release	Dual antitumor immunostimulants	In vitro: BMDCs	[163]
MSNs	Surface	Hexahistidine	Chlorogenic acid	None	Ni scavenging (histidine) Antiinflamatory (Chlorogenic acid)	In vitro: BJ	[164]
Enzymes							
MSNs	In-pore grafting	CA or HPR	None	None	Proof of concept	None	[165]
MSNs	In-pore grafting	CA	None	None	Proof of concept	In vitro: HeLa	[166]
MSNs	Pore loading	β-Galactosidase	None	None	Treatment of Morquio B syndrome	In vitro: N2a	[167]
MSNs	Surface grafting	SOD	None	None	Antioxidant effect (reduction of Reactive Oxygen Species)	In vitro: HeLa	[168]
MSNs	Surface grafting	SOD or GPx	None	None	Antioxidant effect (reduction of Reactive Oxygen Species)	In vitro: HeLa	[169]
MSNs	Surface grafting	Proteasomes	None	None	Anti Tau-protein aggregation	In vitro: HEK-293, HeLa	[170]
UC@MSNs	Surface adsorption	RNAase	Cisplatin prodrug	Electrostatic detachment	Combined anti-protein and chemotherapy	In vitro: HepG2, L929 In vivo: Mice	[171]

Abbreviations: BAMLET: Bovine α-lactalbumin made lethal to tumors; bFGF: basic fibroblast growth factor; BMDCs: murine Bone-marrow derived dendritic cells; BMDMs: murine bone marrow-derived macrophages; BMP-2: Bone morphogenetic protein 2; BMSCs: rat bone mesenchymal stem cells; bMSCs: Murine bone mesenchymal stem cells; CA: Carbonic anhydrase; cAMP: cyclic adenosine monophosphate; MCaSiNs: Calcium Silicate Mesoporous Particles; CpG@OVA: Ovoalbumin loaded CpG oligodeoxynucleotide; DMOHS: Dendritic Mesoporous Organosilica Hollow Spheres; DOX: doxorubicin; EspA: an immunogenic protein from enterohaemorrhagic *Escherichia coli*; GPx: Glutathione peroxidase; HBMSCs: Human bone marrow derived mesenchymal stem cells; HMSNs: Hollow Mesoporous silica Nanoparticles; HRP: Horseradish peroxidase; IgG: Immunoglobulin G; IL2: Interleukin 2; MS-HANs: Mesoporous silica-hydroxyapatite nanoparticles; OGP: Osteogenic growth peptide; ORF2: open reading frame from Porcine Circovirus Type 2; OVA: Chicken Ovoalbumin; RA: Retinoic acid (all trans); RSNs: Rough (non-porous spiky) silica nanoparticles; SOD: Superoxide dismutase; SPIONs: Superparamagnetic Iron Oxide Nanoparticles; UC@MSNs: Mesoporous silica coated up conversion nanoparticles.

The most relevant example was reported by Vallet-Regí and coworkers, who reported the effect of Concanavalin A, a lectin with anticancer and antibacterial properties [116], against murine preosteoblast (MC3T3-E1) and human osteosarcoma (HOS) cells. In this system, doxorubicin (DOX), a common chemotherapeutic, was also incorporated to the MSNs before the polymeric layer which was attached to the silica surface using pH-sensitive linkers. Finally, amide bonds were used to graft the ConA onto the surface. Only when the pH decreased enough was the bis-acetal linker cleaved, enabling the release of DOX. Although not studied, the antiproliferative effect of ConA was ensured when both species were administered simultaneously, showing a synergistic effect higher than the obtained with DOX alone.

In addition to previous examples, another promising anticancer substance is the bovine α-lactalbumin made lethal against tumor cells (BAMLET), which has also arisen as an interesting possibility for silica-based delivery. BAMLET is an emerging nanotherapeutic, which comes out of conjugating bovine α-lactalbumin (BLA) with oleic acid and acts as apoptosis promoter. With this information in mind, Pei et al. designed a nanosystem able to co-deliver a typical cytotoxic drug such as docetaxel (DTX) with BAMLET [117]. Their design was prepared from oleic acid modified MSNs, which upon loading with DTX and treatment with BLA, resulted in the formation of silica particles with a protein coating. This BAMLET corona performed several functions, as it allowed great colloidal stability and biocompatibility together with an effective diffusion barrier for the loaded cytotoxic drug. This protein-coated system was able to exert a relevant proapoptotic effect against several cancerous cell lines such as MCF-7, RBL-2H3 and HeLa, which could be increased in the presence of the cytotoxic drug. In vivo assays on mice showed very promising results, too, as the individuals treated with the DTX-MSN-BAMLET nanosystem showed complete tumor growth arrest and survival after 30 days.

Unlike proteins, whose anticancer effect is promoted by triggering biological effects, peptides can exert anticancer effects throughout two different mechanisms: (1) disruption of membrane’s normal function; and (2) the activation of pro-apoptotic pathways [172]. The first mechanism operates when the peptides are enriched in cationic amino acids: Lysine (Lys, K; amine) Arginine (Arg, R; guanidine) and Histidine (Hys, H; imidazole) and is similar to the effect shown by other polycationic species [173]. Even though this mechanism is not fully understood, recent investigations on nanosystems based on TAT [174] and related cell-membrane-penetrating peptides [175] suggest a membrane-lytic effect too.

Regarding chemical reactivity, peptides bind much more easily to nanoparticles than proteins, as their relatively small size reduces the possibility of linkages on active site surroundings and permit the use of les mild conditions. Nevertheless, despite this advantage, typically polycationic peptides are known to increase the risk of vascular damage like hemolysis and embolisms during trafficking. For these reasons, masking its positive charge could be a relevant strategy in nanomedicine’s design. This was explored by Zhang and coworkers on three different approaches. In the first example, they transformed a custom-targeted cationic peptide K_8_(RGD)_2_ into a negatively charged peptide using citraconic anhydride without affecting its targeting capacity [118]. On later investigations these authors reported the use of glutathione-cleavable anionic coatings for cloaking triphenyl phosphonium modified KLA peptides (KLAKLAKKLAKLAK) [119] and membrane disrupting sequences specific to mitochondria (C-GKGG-DKLAKKLAKLAK) and membranes (C-GRKKRRQRRRPPQ-RGDS) [120]. In these three contributions the authors managed to obtain significantly reduced membrane-lytic effects and demonstrated the importance of enhancing biocompatibility of nanocarriers.

Following another strategy, Zhang’s research group developed a drug delivery system for DOX employing a membrane-targeted therapeutic peptide. Herein, the tryptophan-rich peptide ((RGDWWW)_2_KC) was bound to MSNs through glutathione-mediated disulfide bonds [121], which made it possible to detach the peptide and exert a DNA-intercalant effect. As expected, the best results were obtained when DOX was also present, suggesting a combined effect. Feng et al. also tested this coating strategy for the simultaneous delivery of DOX and the anticancer peptide KLA. Herein, bovine serum albumin (BSA) corona was used as the final capping in this model [122]. This protein had a double effect: it was able to enable a protease/glutathione-mediated intracellular release and established a diffusion barrier for both therapeutic agents. The use of BSA in its wild-type form in the system is an intriguing feature that could result in additional cellular responses when combined with the other multi-apoptotic effects.

Previous examples were developed employing typical coupling processes throughout direct amidation processes. However, more recently cross- and orthogonal linkers has emerged as a powerful strategy for the conjugation of oligopeptides onto particles [176] and between bioactive fragments [177]; although the delivery of biologically active peptides is still based on the reversible bindings. In contrast to precedent examples, peptide pore loading throughout threading is also a highly convenient strategy for delivery as it makes it possible tothe incorporation of additional peptides onto particles’ surfaces. This strategy was explored by Martnez-Máñez’s group, who reported the use of polylysine as coatings on the C9h (YVETLDDIFEQWAHSEDLK) loaded pro-apoptotic peptide [123]. Herein, this polylysine layer performed two different roles, whereby its cationic nature (1) facilitated cellular uptake and (2) prevented C9h peptide degradation and leakage. The encapsulated peptide showed a better therapeutic profile when delivered from MSNs rather than in its free form. Unfortunately, tested therapies employing this peptide quickly reached a maximal effect far away from clinic desirable efficiencies, underlining the limited anticancer effects of peptides and the need for combining therapies to produce relevant results.

Other examples on proapoptotic peptides delivery were reported by Cao et al. [124] and Rahmani et al. [125], who studied the behavior of Hylin a1 peptide (IFGAILPLALGALKNLIK) and the Cathepsin D inhibitor peptide pepstatin A, respectively. In the first example, the authors tuned the mesoporous environment to facilitate threading and enable pH-dependent releases. Moreover, they also elongated the peptide strand to include an additional RGD recognition sequence [124]. In vitro studies for this system revealed that peptide encapsulation reduce hemolysis showed by the free peptide, while maintained a potent cytotoxic effect against HeLa and Hep2 cells. In the second example, the authors developed a silica-based formulation to increase peptide dosage. They evaluated both mesoporous and hollow mesoporous materials to determine the best delivery profile. Unexpectedly, these authors discovered two diverging behaviors: HMSNs were able to load less peptide than typical large-pore MSNs. However, they also showed a more sustained release (longer therapeutic effect) compared to pore-expended MSNs, which is of high interest for future developments.

In addition to the previous examples, anticancer research interests have also focused on the repurposing of bioactive substances to help fighting the disease. For example, Ferreira Soares and coworkers studied the effect of alamandine, a heptapeptide related to angiotensin [126]. According to their results, this peptide showed an improved therapeutic effect against tumoral 4T1 (murine breast cancer) and A549 (human pulmonary cancer) when loaded within the mesopores and a very low cytotoxic effect on HEK-293 cells probably consequence of the protective role exerted by the silica carrier. Other interesting examples were reported by Ramya et al. [127] and Lin et al. [128], who respectively employed Fabatin, a phytodefensin peptide obtained from the seeds of *Vicia faba*, and Mellitin, a water-soluble cationic peptide present on bee venom, in anticancer nanodevices. In the first example, Ramya et al. loaded the Fabatin peptide (LLGRCKVKSNRFNGPCLTDTHCSTVCRGEGYKGGDCHGLRRRCMCLC) into unmodified MSNs [127]. This simple formulation was able to improve cytotoxicity of free Fabatin and drop viability of triple negative breast cancer MDA-MB-231 cells to values close to those obtained with DOX. These values were also maintained when the Fabatin-loaded nanosystem was tested in mice. Indeed, apoptosis levels on other tissues were reduced 5-fold compared with the values for chemotherapy, highlighting the potential of peptide nanoformulations. In the second example, Mellitin peptide (GIGAVLKVLTTGLPALISWIKRKRQQ) was delivered from MSNs through a sequential magnetic-enzymatic double triggering stimulus-responsive nanosystem. When the peptide was retained within the pores, great hemocompatibility and low cytotoxicity were obtained. However, when the peptide was released, viability reductions above 80% were possible to obtain in PANC-1 cells. Evaluation in mice with the combined peptide and thermal therapies, the nanosystem was able to promote tumor growth reduction of about 75% in comparison with the control. Considering that both Fabatin and Mellitin were the only therapeutic agents within these nanosystems, it seems clear that repurposing peptides with the aid of nanotechnology could open the door to new generation alternatives to conventional chemotherapies.

### 3.2. Antibacterial Therapies

In addition to the anticancer examples, proteins may also exert antibacterial effects too [178]. The widely known lysozyme, a 14.4 kDa enzyme, has been widely employed to destroy Gram-positive bacterial walls by cleaving the −1,4 bonds between *N*-acetylmuramic acid and *N*-acetyl-D-glucosamine. In the first example reported on the delivery of antibacterial proteins with silica, negatively charged MCM-41 MSNs were employed to electrostatically bind lysozyme units [129]. The resulting enzymatic activity on the bacterial microenvironment made it possible to destroy both Gram-positive and Gram-negative bacteria. Subsequent investigations aimed at increasing this bactericidal effect led to the development of different nanosystems. Among them, dendritic pore MSNs [130] and a set of fancy, rough, non-porous spiky nanoparticles [131] with very interesting protective features were also employed as support for lysozyme. The comparison between different models suggests that the antibacterial effect of lysozyme is mainly achieved when quick releases occur, although in the case of the spiky particles they seem to exert an additional unknown effect. Unfortunately, all these systems were tested only against planktonic bacteria, which are the most sensitive and less pathogenic state. In a more recent contribution, Ye and coworkers investigated if enlarged pore, hollow MSNs with large loading capacities (up to 350 mg·g^−1^) were able to destroy preformed biofilms [132]. In their work the authors found an activity threshold (400 g·mL^−1^) for lysozyme over which no added therapeutic effect could be obtained. In this case, the use of the carrier raised was able to obtain more prolonged effects than free lysozyme due to a more sustained release.

In addition to lysozyme, Concanavalin A (ConA) has also demonstrated certain antibacterial effects [133]. Vallet-Regí and coworkers profited from the antibiotic properties of such lectin to prepare an antibacterial nanosystem in combination with levofloxacin. Their model, based on the chemical grafting on ConA at the final step onto carboxylate-modified MCM-41 MSNs made it possible to successfully prepare a nanodevice able to deliver these two antibiotic species alike. As a result, the dual action of this nanosystem made it possible to destroy *E. coli* biofilms at a concentration of 10 mg MSNs per mL, demonstrating that combination therapy outperforms single-agent therapies in antibacterial treatments too.

Along this line, Bactofencin A also displayed an intriguing antibacterial action when delivered with SBA-15 particles. In the contribution by Hudson and coworkers a clear antibacterial effect of such protein was demonstrated against *S. aureus* [134]. In this case, the system did not include diffusion barriers, which led to a rapid outflow of the protein and thus, an almost total release within 5 h. As a result, the remaining bacteria were able to reform the biofilm quickly and return to its pathogenic state.

To conclude with the silica-based delivery of antibacterial proteins, it is noteworthy to highlight, too, the potential of human proteins for antibacterial therapies on other species. Along this line, Marcelino-Pérez et al. reported the effect of a recombinant human β-defensin 2 onto mesoporous silica nanoparticles as phytosanitary against *Clavibacter michiganensis* subspecies *michiganensis* [135]. In this work the authors obtained the human β-defensin 2 upon transfection of the engineered plasmid in *E. coli*. and loaded the protein directly into pore-enlarged silica particles. The nanosystem proved to be highly efficient against *C. michiganensis* as it was able to reduce the number of colony-forming units in plants by 10-fold. Moreover, the use of a human protein as phytosanitary is expected to have negligible effect in public health due to its human origin.

Peptides are another research line with broader potential to develop antibacterial treatments, as they promise to overcome bacterial resistance throughout non-conventional mechanisms. In fact, the integration of such antimicrobial peptides into nanocarriers has also received attention during last years [179,180]. The first examples on the topic were focused mainly on methodological aspects and on peptides able to undergo membrane-disrupting effects. However, as introduced before, recent contributions also focus on therapeutically active peptides. For example, the studies by Braun et al. on the optimal loading and particle composition (non-porous, calcinated mesoporous, and amino-capped mesoporous silicas) employed the LL37 (LLGDFFRKSKEKIGKEFKRIVQRIKDFLRNLVPRTES) antibacterial peptide [136]. The most negatively charged nanoparticles (calcined MSNs) showed the best loading profile and thus antibacterial effect against *E. coli*. The authors suggested that this effect was consequence of an optimal charge balance between both counterparts. Moreover, the appropriate protective environment of porous silica also helped in decreasing hemo- and proteolysis caused by LL37. In a later work, these authors also investigated the effect of the porous structure. By comparing enlarged pore HMSNs with typical sized-MSNs they discovered that the HMSNs showed a higher antibiotic profile consequence of a more continuous release [181], in agreement with data reported by Rahmani et al. [125]. This LL37 peptide was also successfully employed as antibiotic on silica carriers with virus-like topographies. In this contribution, reported by Häffner et al. [137], spiky nanoparticles had a crucial role, they were able to open bacterial and favor internalization of LL37 into bacteria. As a result, the complete nanosystem was able to reduce *E. coli* colonies in more than 80%, and also prevent the formation of new biofilms and bacterial aggregates.

Another promising research line for antibacterial nanodevices is the treatment of infected cells. Along this line, Tenland et al. employed MSNs to deliver to infected macrophages the NZX [138] (GFGCNGPWSEDDIQCHNHCKSIKGYKGGYCARGGFVCKCY) anti-tuberculosis peptide [182]. This MSN loaded nanosystem proved to internalized by macrophages and produce an effective peptide release able to destroy infected mycobacterium without significantly affecting host macrophages. Moreover, this particular formulation made it possible to obtain a longer therapeutic effects than free. The authors also suggested a possible operating mechanism based on the accumulation of nanocarriers within intracellular vacuoles which acted as drug reservoirs by the prevention of peptide digestion within the silica matrix.

In recent years, other emerging antibacterial peptides have also been nanoformulated using porous nanosilica as carriers. For example, Ma et al. employed an ovotransferrin-derived peptide to create a multipurpose nanotherapeutic able to act as antibiotic drug delivery system [139]. These authors demonstrated that the surface functionalization of MSNs with the OVTp12 (AGLAPYKLKPIA) peptide alone was able to induce an interesting antibiotic effect against *E. coli*, although this effect was clearly enhanced when the mesopores were loaded with gentamicin. In vivo studies of this system showed very low hemolysis rates (<10%) and a potent anti-infective effect in mice, reaching up to ca. 70% survival of individuals after 10 days.

Another interesting example on combination therapy was reported by Alharthi et al. who studied the antibiotic effect of different combined Sortase A inhibitors [183] (*trans*-Chalcone, Curcumin, Quercetin and Berberine chloride) together with polycationic antimicrobial peptides: Pexiganan (GIGKFLKKAKKFGKAFVKILKK-NH_2_), Indolicidin (ILPWKWPWWPWRR-NH_2_) and Mastoganan (INLKILARLAKKIL-NH_2_) [140]. All examples employed inhibitor-loaded raw or phosphonate-modified MSNs which were further coated with the corresponding peptide throughout electrostatic deposition. As expected, obtained results showed that all combinations of inhibitors and peptides displayed different degrees of antibacterial effects. However, for certain combinations, the overall effect of combined therapy clearly improved the minimum inhibitory concentration values obtained for single-loaded therapeutics, suggesting that the repurposing of low-effect but cheap therapeutics could have importance on the development of new generation of nanomedicines and help in the fight against bacterial resistance.

In another interesting contribution to the topic, Zhang et al. reported the use of SiO_2_ nanoparticles to create skin-inspired fibrous membranes with wound healing properties. Herein, silica particles were chemically modified to behave as the outermost hydrophobic coating of a multilayered skin-like structures [184]. Membranes were assembled throughout sequential deposition of hydrophilic and hydrophobicity components such as poly(vinyl alcohol) and poly(vinylidene fluoride) along four layers. Additionally, a quinolone antibiotic (Ciprofloxacin), together with an antioxidant (Astaxanthin), were embedded within the structure to promote the proliferation of new fibroblasts and prevent bacterial growth. As a result, this silica-containing composite was able to be implanted onto massive wounds and herein induce new tissue regeneration and complete wound healing within 2 weeks. In addition to skin regeneration, silicon-based ceramics have also demonstrated antibacterial features. As demonstrated by He and coworkers [185], polypropylene-Si_3_N_4_ composites showed bacteriostatic effects against *E. coli* and *S. aureus* in non-woven filtering materials such as facemasks. Despite these examples not containing therapeutic biopolymers, they show that silicon-containing ceramics are also highly convenient materials for the preparation of a broad number of healthcare materials.

### 3.3. Osteogenic Therapies

Growth factors, as promoters of cellular proliferation and differentiation are crucial for the development of functional vasculature, bones and wound healing. During the last few years, many studies on aging and frailty have pointed out that osteoporosis and related bone diseases could be successfully treated with both hormonal or nanomedical therapies, with this opening the way to a new generation of bone healing nanodevices [186,187]. This topic, pioneered by Zhang et al. more than a decade ago [141], has now become a hot research area in the uses of silica in nanomedicine. This first contribution, in which ad hoc MSNs were designed to host the 18 kDa basic fibroblast growth factor (bFGF) [141] was quickly followed by two interesting contributions by Gan et al., who studied two complimentary strategies to deliver the bone morphogenetic protein 2 (BMP-2). In the first contribution, these authors explored the behavior of dexamethasone-loaded, chitosan-coated MSNs to deliver such proteins attached onto particles’ surfaces [142]. whereas in the second model, a pore protecting strategy was explored. In this work, the SBA-15 mesopores were capped with iron oxide nanoparticles [143] to reduce the excessive osteogenesis obtained in their first example.

Some peptides also have anabolic properties, for which reason they have also been employed as payloads in bone repairing nanodevices, especially those that focus on local tissue regeneration. Along this line, Esbrit and Vallet-Regí studied bone remineralization triggered by Osteostatin, a peptide derived from the parathyroid hormone-related protein (PTHrP_107-111_) [144]. In their two contributions these authors employed SBA-15 matrices to ensure peptide preservation and sustained release. In fact, the longer-lasting effect of loading allowed these materials to induce bone regeneration in both cavitary defects [145] and osteoporotic defects [146] successfully, as demonstrated by the significant increments on bone mass obtained. Similar strategies were also reported by Mendes et al. [147] and He and coworkers [148], who reported the successful combination of therapeutic peptides with other chemical species such as (1) the Osteogenic Growth Peptide (OGP, ALKRQGRTLYGFGG) and apatite onto SBA-15 matrices and (2) the BMP-2 derived peptide (KIPKASSVPTELSAISTLYL), together with the synthetic glucocorticoid dexamethasone employing MCM-41 nanoparticles.

In more recent studies, the combination of Osteostatin with CaP containing bioactive glasses [188,189] has emerged as a powerful tool for the development of new bone therapies promoted by local delivery of calcium and phosphorus. Additionally, in recent examples, new elements have been incorporated to bioactive glasses to enhance their therapeutic effects and properties. For instance, Jiménez-Holguín et al. reported the use of Cu for bone mineralization with additional antimicrobial effect [190,191]. Mutreja et al., also reported the use of porous Sr-doped calcium silicate glasses to deliver the GL13K (GKIIKLKASLKLL) peptide. This peptide, derived from the human salivary parotid secretory protein, was chosen because its antimicrobial and osteogenic properties (reduces osteoclast differentiation). As a result, peptide delivery was able to promote bone mineralization in combination with an anti-infective effect against *Staphylococcus aureus* [149]. Data also pointed out effective increments of typical proteins associated with bone formation, demonstrating the potential of this multipurpose peptide for treating and preventing bone diseases.

### 3.4. Immunostimulant Proteins and Peptides

Immunotherapy is one of the most promising strategies for improving cancer therapies. The general idea is to recruit patient’s immune system to attack tumors too. Nevertheless, any indiscriminate or uncontrolled activations may have disastrous consequences on patients’ health. For this reason, the scientific community has shown growing interest in the development of novel therapeutics for local applications. In cancer, the enhanced retention and permeation effect (EPR) may help achieving this goal by favoring accumulation of immune stimulants within tumoral area. Then, in an ideal case, the local activation of the immune system would produce a response able to recognize and destroy cancerous cells in a vaccination-like strategy. For interested readers, a very recent review focused on the most convenient materials, aspects and limitations of this strategy [192]. In this section, we also remark that MSN-based therapies have come to be of great importance because of their features as immune adjutants [193,194,195].

According to published examples, there are several strategies that allow the incorporation of antibodies into silica carriers. In addition to already-discussed surface deposition and grafting and pore loading, new materials have been also developed for such purposes. One of these examples, reported by Lim et al., unique hollow MSNs showing unusual perforations were employed to load immunoglobulin (IgG) to HeLa cells [150]. Unfortunately, despite effective intracellular delivery, no immune response was achieved. Similarly, Niu et al. proposed different core–shell nanoassemblies between solid silica particles of different sizes. The resulting rough surfaces were able to host proteins within interparticle voids [151]. In particular, these authors reported nanostructures with different roughness (14, 21 and 38 nm) which could host and preserve different proteins such as Cyt c, monoclonal rabbit immunoglobulin and bigger antibody-containing proteins (HRP-linked anti-rabbit IgG antibody). In vitro studies with 38 nm hydrophobically modified RSNs showed effective deliveries of anti-pAkt antibodies into MCF-7 breast cancer cells yielding a considerable impact on cell proliferation and the downregulation of the anti-apoptotic Bcl-2 protein.

Apart from those, silica nanoparticles have also been employed for the delivery of immunostimulant proteins. Along this line, Wang, Ito, Tsuji, and colleagues, started a very prolific line of research on evaluating the stimulant effect of Hollow MSNs loaded with a chicken ovalbumin (OVA) in vivo [152]. The treatment of mice with this nanosystem produced a significant immune activation through increases of CD4+ and CD8+ T-cell populations, on the upregulation of Th1 and Th2 cytokines and of the increase in immunoglobulin antibody levels [153]. Apart from OVA, these authors also determined that employed MSNs were able to exert an immune stimulant effect, which enhanced the overall effect. Similar results were also reported by Yang et al., who reported the delivery of OVA in combination with tumor antigens from B16F10 cells. To ensure adequate and simultaneous release of OVA and antigens, rapidly degradable multi-shelled dendritic mesoporous organosilica hollow spheres (DMOHS) were chosen [154]. As a result, upregulation of CD4+ and CD8+ T-cells in combination with overexpression of Th1, interleukin-12 (IL12), gamma interferon (γ-IFN), and alpha tumor necrosis factor (α-TNF) were produced. In addition to those examples, Cha et al. also evolved the strategy by including an additional immunostimulant substance into a silica-based carrier. In their experiment, OVA was delivered in combination with an immunological danger signal, one agonist of the Toll-like Receptor 9 (TLR9). Both species were successively loaded into large pore MSNs (20–30 nm) [155]. As a result, this system induced a similar response that those described in previous system but in a much higher scale.

Potential anticancer vaccines have also been reported employing interleukin 2 (IL2) as payload. In the work by Kong et al., IL2 was delivered in combination with all-trans retinoic acid and doxorubicin. The system was completed employing biodegradable hollow mesoporous silica nanoparticles and a lipid coating to prevent leakage and to favor biocompatibility [156]. This combination of drugs produced a general antitumor effect in mice and a significant reduction on melanoma metastasis potential, although complete tumor regression could not be achieved.

Apart from anticancer vaccination, delivery of immunoproteins is also of interest for the development of nanomedicines against other diseases. For example, Park et al. reported an Interleukin 13 (IL13)-loaded nanosystem for the treatment of autoimmune encephalomyelitis [157]. This system employed large pore MSNs (ca. 63 nm) to load the protein within the silica matrix. The intracellular delivery of IL13 induced the differentiation of bone marrow-derived macrophages to M2 macrophages, as demonstrated by the increment of CD11b, CD206 markers and the F4/80 antibody. In vivo, this nanosystem improved prognosis of the experimental autoimmune encephalomyelitis after only two intranasal injections.

In addition to previous examples non-viral-based vaccination could be also targeted for other common diseases [196,197]. The main idea is based on the delivery of antigens to immune cells without triggering a whole immune response. Along this line, several examples employing silica nanoparticles could be found in the literature. For example, Guo et al. employed HMSNs as carrier for the porcine circovirus type 2 (ORF2) protein [158]. In this nanosystem the protein was directly adsorbed onto raw nanoparticles without an additional protective layer. Despite the possible degradation, this nanoformulation achieved effective immunological activation in mice, as demonstrated by the overexpression of CD4, CD8, γ-IFN and Th1 immunoproteins. During the following years, other non-viral-based nanovaccines against (1) *Schistosoma mansoni,* by employing homogenates from the parasite [159]; (2) porcine enzootic pneumonia, by using a recombinant HSP70 antigen fragment (HSP70_212-600_) from *Mycoplasma hyopneumoniae* [160]; (3) enterohaemorrhagic *E. coli*, by using a recombinant fragment of filamentous EspA protein [161]; and (4) *Paracoccidioides brasiliensis* employing the rPb27 antigen [162] were developed following similar strategies. Therapeutically, all of these nanosystems were able to immunize mice against pathogens of different nature reaching pharmacological profiles similar to those reported for free antigens and other pharmaceutical formulations with much lower macrophage’s internalization rates.

Despite their activity usually being lower than those of proteins, peptides could also be employed to induce immune responses. This is of especial interest in the case of peptides that retain biological aspects from their parent proteins. For example, Xie et al. reported the use of hollow mesoporous silica carriers to deliver two melanoma-derived antigen peptides: the hydrophobic TRP2_180-188_ (SVYDFFVWL) and the hydrophilic HGP100_25-33_ (KVPRNQWL) peptides [163]. In order to load both peptides, the authors modified the nanocarrier with amino groups at the internal cavity and with carboxylate groups at the mesopores to enable a sequential differential double loading. To preserve both peptides and to provide enough colloidal stability, the nanosystem was further coated with a lipid bilayer in which there was incorporated an additional therapeutic compound, the toll-like receptor 4 (TLR4) agonist monophosphoryl lipid A. According to reported data, this nanosystem promoted immune cell maturation as evidenced by the upregulation of CD86, α-TNF, γ-IFN, IL12, and IL4 proteins. In vivo, vaccinated animals showed a promising tumor growth reduction on melanoma tumors together with a lower development of metastatic lymph nodes, being one few nanosystems able to reach such anticancer efficacy.

In addition to the delivery of immunostimulant compounds, recent investigations have also focused on recruiting immune cells to create advanced delivery systems. Such is the case of the article reported by Lei and Tang, who employed mesoporous silica microparticles to develop a T-cell responsive drug delivery system [198]. This system was based on a double strand DNA able to block mesopores [39]. To achieve so, one of the strands was linked to the silica while the complementary strand was bound to an anti-CD3 antibody. As a result, the hybridized DNA was able to keep Gemcitabine loaded within the pores in the absence of T-cells. However, in the presence of CD3 presenting T-cells, antibody-antigen affinity was able to dehybridize the double-stranded DNA-producing pore opening and drug outflux. The results obtained with this proof of concept suggested that this drug delivery strategy could be of interest for the development of nanoparticle-based therapies for certain lymphomas.

Apart from the presented examples aimed at the activation of the immune system to treat diseases, combinations of peptides and MSNs have been also employed to reduce immune responses. The example reported by Wang et al. was able to trap Nickel and thus reduce allergy levels [164]. To do so, the authors profited from the extraordinary coordination ability of histidine groups towards Ni cations. Through adequate chemical modification, hexahistidine peptides were grafted onto MSNs, yielding a nanosystem able to scavenge Ni and reduce allergic responses to this metal. In addition to the peptide, this nanosystem was also loaded with chlorogenic acid, an emerging cosmetic component [199] with claimed anti-inflammatory and wound healing properties in order to enhance the antiallergic performance.

### 3.5. Enzymatic Therapy

The primary cause of metabolic disorders is the dysregulation of enzymatic functions. This situation may lead to toxic syndromes if it remains untreated. In cancer, the dysregulation of homeostasis is also a side effect of the aberrant enzyme expression. For both reasons, enzymatic therapy has also become a promising field of research in nanomedicine, as well as when considering the lability of enzymes. Current enzymatic therapies consist of repetitive injections of enzymes, which usually leads to low-adherence treatments. In order to improve current treatments, enzymes must be modified to increase their lifespan [200]. Such are the cases of collagenase capsules, Villegas et al. [201] and anticancer p53 nanocapsules reported by Zhao et al. [202], whose degradable protective shells preserve unaltered the enzymatic function during trafficking.

However, when dealing with porous silica nanomaterials, the enzymes could be located either at the surface or within the mesopores. This second strategy is preferable for preventing enzymatic activity loss, although at the expense of being a considerable diffusion barrier for bigger substrates. Governed by simplicity, most of reported MSN-based carriers locate the proteins at the surface, as this facilitates the flux of both substrates and products. One of the most widely employed enzymes in the literature is Carbonic Anhydrase (CA), which, despite having no therapeutic impact, is widely employed as a model [165,166], especially due to the ease of measuring carbon dioxide and bicarbonate.

Apart from previous proofs of concept, MSNs have also successfully been employed as support and carriers for therapeutic enzymes, especially those aimed at the reduction of toxic syndromes. For instance, Xu et al. reported a promising model to reduce the severity of Morquio type B syndrome [167]. This disease occurs when Galactosidase enzymes fail to break the glycoside bond, thus causing bioaccumulation of oligosaccharides. In their model, the authors hosted the enzyme within the mesopores rather than on particles’ surfaces, increasing the lifespan of the enzyme due to the improved protection offered by the silica matrix. It is important to remark that the small sizes and good solubility of substrate oligosaccharides made it possible to implement this configuration due to the facility of diffusion to and from mesopores. In this example ultra-large pore MSNs were required to host the 119 kDa enzyme; being core-cone shaped MSNs with dahlia-like mesopores the chosen structures. These particles were effectively internalized into N2a cells and once up taken, were able to maintain its catalytic activity much longer than the free enzyme.

Superoxide dismutase (SOD) and glutathione peroxidase (GPx) are two antioxidant enzymes widely employed to reduce Reactive Oxygen Species (ROS) intracellularly. Nevertheless, the different nature of ROS requires from nanosystems the ability to adequately expose enzymes to exert an effective antioxidant function. The system reported by Mou and colleagues for SOD delivery employed a nitrilotriacetic acid (Ni-NTA) modified silica to bind a TAT-enzyme conjugates throughout Ni^+2^ chelation [168]. To prevent extracellular ROS depletion promoted by Paraquat, the authors denaturalized the enzyme with urea, a well-known superoxide anion generator. However, when the particles were internalized by HeLa cells, the enzyme was refolded into its active state and produced the desired antioxidant effect. More recently, Mou and coworkers studied the effect of delivering two antioxidant enzymes. For so they prepared two differently loaded nanodevices—one with SOD and the other with GPx [169]. In their investigations, these authors discovered a complementing synergistic effect with this combination, where the co-delivery of both antioxidant proteins enhanced the impact of individual therapies.

Focusing on Alzheimer’s disease, Han et al. provided an interesting example able to prevent aggregation of tau proteins through the introduction of human proteasomes [170]. These authors employed Mou’s strategy to bind histidine moieties from active human 26S proteasomes isolated from the HEK293 cell line. The resulting system demonstrated that these enzymes could be distributed without significantly increases of proteotoxicities. Indeed, the authors asserted that their approach might be of interest for delivering proteins throughout impermeable barriers, although this system did not have potential to prevent Alzheimer’s condition in vivo. In fact, for in vivo applications, all nanoparticle-based drug delivery system must be able to cross into the brain through biological barriers (blood–brain, blood–brain tumor, nose-to-brain, etc.), which require specific strategies. For an in-depth discussion on the topic, please refer to the review presented in [203].

In conclusion, enzymatic therapy may also be employed to treat cancer diseases. This was demonstrated by Teng et al., who developed a nanosystem able to deliver RNAase intracellularly. This enzyme was able to disrupt the abnormal expression of proteins within cancerous cells and thus reduce its aggressivity [171]. In their model, up-conversion nanoparticles coated with a mesoporous silica layer (UC@MSNs) were loaded with a cisplatin prodrug and coated with the RNAase to promote downregulation of proteins. As a result, the combined effect of both species boosted HepG2 apoptosis levels to almost 50% according to flow cytometry data. In vivo evaluation in mice of this nanosystem permitted stopping tumor growth and maintained tumor volume constant for almost two weeks, although these tumors could not reach complete remission.

## 4. Conclusions

As mentioned throughout the manuscript, the capacity and versatility of nanometric porous silica materials to develop potential therapeutic systems is enormous. This is partially a consequence of their feasibility in the development of novel strategies in nanomedical approaches. Moreover, silica offers one of the simplest methods to develop nanosystems for combination therapies, which is clearly demonstrated by the great number of systems with therapeutic potential that have been reported up to date. In addition to the delivery of oligonucleotides, peptides and proteins reviewed herein, silica particles also offer many other possibilities for creating nanotherapeutics such as targeted nanomedicines and stimuli responsive systems. However, as a consequence of their easy tunable porous structure, it is also possible to house a significant number of small and medium-sized molecules of practically any known chemical nature within the particles’ matrices.

However, the use of mesoporous silica for nanomedical developments also has severe limitations. For instance, even though they are biodegradable, their slow dissolution rates and sturdiness cause them to accumulate relatively easily within tissues and during trafficking. Indeed, if inadequate surface modifications are performed, these particles show a certain tendency to aggregate and therefore produce thrombosis and embolisms. On the other hand, the delivery of functional biomacromolecules with silica-based transporters presents important limitations too, the main one being the possible immune response of patients. In fact, many nanomedical designs fail when they are tested in immunocompetent models. Fortunately, the technology needed to overcome these problems is known and although the use of biopolymers such as nucleotides, proteins and peptides can alleviate these side effects to a large extent, this technological development is still in its infancy.

In addition to previous issues, the development of oligonucleotide drug delivery technologies also led to other limitations: the use of positively charged supports and polymers. This reduces overall biocompatibility in comparison with neutral or anionic systems due to the higher hemolysis rates. Although some research groups have tried to solve this problem by using fewer positive charges or by masking polycations, the truth is that it seems to be a rather difficult problem to solve. Proof of this is that all new-generation COVID-19 vaccines have been developed using liposome technologies, which is completely different from solutions offered by oligonucleotide delivery with silica nanoparticles. Similarly, in the case of protein and enzyme delivery, there also are important limitations to be solved. In the case of in-pore loading, it is necessary to adapt pore sizes to proteins sizes and shapes while on surface location technologies that avoid the formations of protein coronas and avoid the blockage of active sites remain unsolved.

Regardless of these drawbacks, mesoporous silica nanotechnology also offers significant advantages. Among them, the main advance is the ease of developing combined therapies, which to our opinion are the future of anticancer therapies, as they have proved to be the most successful strategies to achieve complete remission of tumors. Moreover, and as outlined in this review, silica also offers interesting possibilities for the development of topical and oral therapeutics that do not require systemic injections and may favor translation into clinical practice. Even though silica nanotechnology does not reach clinical practice, we still foresee a brilliant future for this technology. In our opinion, silica has already been established as an important source for future nanomedical technologies and will remain so in the near future.

## Figures and Tables

**Figure 1 pharmaceutics-15-00351-f001:**
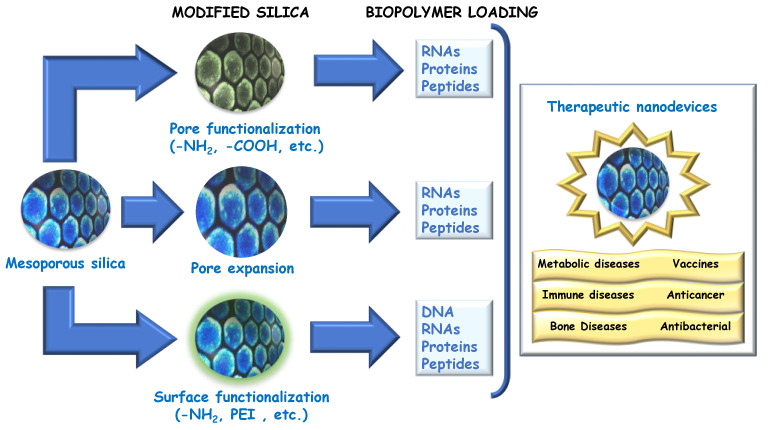
Workflow for building silica-based therapeutics for reviewed diseases. Different bioactive macromolecules could be incorporated onto the silica and efficiently dosed by properly engineering particles porous structure and surfaces.

**Figure 2 pharmaceutics-15-00351-f002:**
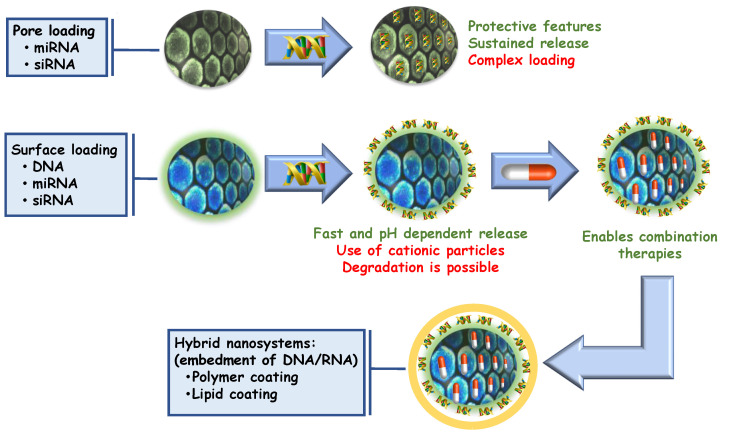
General strategies for nucleic acid drug delivery employing mesoporous silica nanoparticles.

**Figure 3 pharmaceutics-15-00351-f003:**
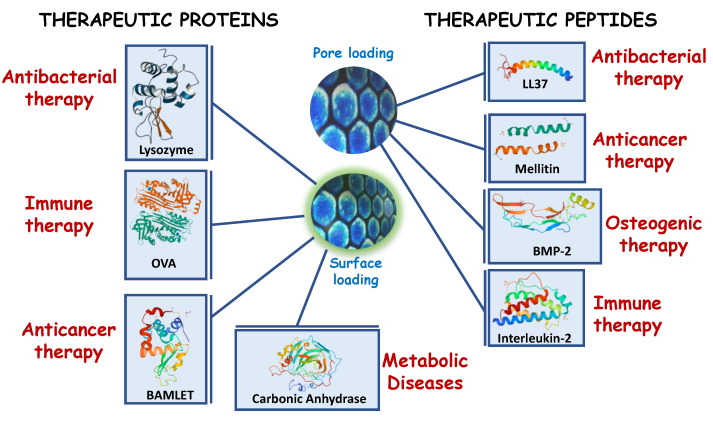
Delivery of proteins and peptides with mesoporous silica nanoparticles makes it possible to create nanotherapeutics for a wide range of diseases.

## Data Availability

Not applicable.

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
