# Peer review of "Delivery of Therapeutic Biopolymers Employing Silica-Based Nanosystems"

_pharmaceutics, 2023, doi:10.3390/pharmaceutics15020351_

Round 1

Reviewer 1 Report

The Review Article " Delivery of Therapeutic Biopolymers employing Silica-based Nanosystems" intends to focus on the most recent advances reported on the use of biogenic polymeric compounds (proteins, nucleotides, and polysaccharides). The general idea of this review article is innovative and of high interest, in my opinion. Below I comments that could transform the article into a better format :

- Figure 1 can be more specific and scientific to present the goal of the review, please revise it

-please remove the “dot” at the end of the title

-I have some concerns with the safety of Si NPs for use in body, if authors agree please address this concern in challenges

- There are many studies investigating the importance of the topic, Please add these references to your introduction and discussion parts of the manuscript and compare and bold your study novelty: https://doi.org/10.2147/IJN.S243223, https://doi.org/10.1016/j.apsusc.2022.155290, https://doi.org/10.3390/polym14142952, https://doi.org/10.1016/j.molstruc.2022.132922

- There are some spelling errors and logic problems in the text that need attention. Moreover, the typos in the manuscript need to be double-checked.

- The conclusion that needs to be upgraded may be Including a discussion part with the cost, and possible side effects or limitations to increase the impact of the paper.

Author Response

The Review Article " Delivery of Therapeutic Biopolymers employing Silica-based Nanosystems" intends to focus on the most recent advances reported on the use of biogenic polymeric compounds (proteins, nucleotides, and polysaccharides). The general idea of this review article is innovative and of high interest, in my opinion. Below I comments that could transform the article into a better format:

We kindly appreciate your support and all valuable comments to improve the overall quality of the manuscript. This updated version has gone through an extensive language editing process to improve readability. We expect you to find this version of the manuscript more pleasing. You can find the track changes version below.

 - Figure 1 can be more specific and scientific to present the goal of the review, please revise it

We agree with this comment and have expanded the figure to be more specific. We hope this new image pleases you.

-please remove the “dot” at the end of the title

This issue has been corrected as suggested.

-I have some concerns with the safety of Si NPs for use in body, if authors agree please address this concern in challenges

As accurately highlighted by reviewer, silica nanoparticles are not currently available to clinical practice. However, we believe there is broad evidence that these particles could be used for in vivo applications. To justify this assumption, we have included a paragraph and several references on the topic:

Among all nanosystems available, porous silica is one of the most widely employed for the development of potential nanotherapeutics, even though mesoporous silica is still not approved for use in clinical practice. [8,9] Nevertheless, SiO2 is generally considered a safe material by the FDA and widely employed in food industry and even structural basis for certain living organisms. [10] Indeed, there are many examples in which biosafety and biocompatibility of silicas have been demonstrated, [11–13] which suggest that silica may be used either alone or as component for the development of new nanotherapeutics.

- There are many studies investigating the importance of the topic, please add these references to your introduction and discussion parts of the manuscript and compare and bold your study novelty: https://doi.org/10.2147/IJN.S243223, https://doi.org/10.1016/j.apsusc.2022.155290,  , https://doi.org/10.3390/polym14142952, https://doi.org/10.1016/j.molstruc.2022.132922

We appreciate the effort made by the reviewer to broaden the scope of the revision. Even though we believe some of these references do not precisely match the content of the topic, we understand that they could be of interest too. Please find the references in the following paragraphs:

Regarding nanoparticles, there is an overwhelming number of them. Paying attention to their different chemical nature, shape, size, and topology they could be classified in many ways. [5,6]

In another interesting contribution to the topic, Zhang et al. reported the use of SiO2 nanoparticles to create skin-inspired fibrous membranes with wound healing properties. Herein, silica particles were chemically modified to behave as the outermost hydrophobic coating of a multilayered skin-like structures. [155] Membranes were assembled throughout sequential deposition of hydrophilic and hydrophobicity com-ponents such as poly(vinyl alcohol) and poly(vinylidene fluoride) along four layers. Additionally, a quinolone antibiotic (Ciprofloxacin) together with an antioxidant (Astaxanthin) were embedded within the structure to promote the proliferation of new fibroblasts and prevent bacterial growth. As a result, this silica-containing composite was able to be implanted onto massive wounds and herein induce new tissue regeneration and complete wound healing within 2 weeks. In addition to skin regeneration, silicon-based ceramics have also demonstrated antibacterial features. As demonstrated by He and coworkers,[156] polypropylene-Si3N4 composites showed bacteriostatic effects against E. Coli and S. Aureus in non-woven filtering materials such as facemasks. Despite these examples do not contain therapeutic biopolymers they do show that silicon-containing ceramics are too highly convenient materials for the preparation of a broad number of healthcare materials

Nevertheless, despite this nanosystem may have potential to prevent Alzheimer’s condition in vitro, the truth is that in vivo all nanoparticle-based drug delivery system must be able to cross into the brain through biological barriers (blood–brain, blood–brain tumor, nose-to-brain, etc.) which require from specific strategies. For an in-depth discussion on the topic, please refer to the following revision. [204]

- There are some spelling errors and logic problems in the text that need attention. Moreover, the typos in the manuscript need to be double-checked.

As commented before, we have submitted the manuscript to an extensive language edition in which we expect most of the mistakes and typos have been removed.

 - The conclusion that needs to be upgraded may be Including a discussion part with the cost, and possible side effects or limitations to increase the impact of the paper.

We have rewritten the conclusion section considering this valuable comment. We hope this new text will please you.

As mentioned throughout the manuscript, the capacity and versatility of nanometric porous silica materials to develop potential therapeutic systems is directly enormous. This is partially consequence of its feasibility to develop novel strategies with interest for nanomedical approaches. Moreover, silica offers one of the simplest methods to develop nanosystems for combination therapies; which is clearly demonstrated by the great number of systems with therapeutic potential that have been reported up to date. In addition to the delivery of oligonucleotides, peptides and proteins reviewed herein, silica particles also offer many other possibilities to create nanotherapeutics such as targeted nanomedicines and stimuli responsive systems. But also, and consequence of their easy tunable porous structure, it permits to house within particles’ matrices a important number of small and medium-sized molecules of practically any chemical nature known.

However, the use of mesoporous silica for nanomedical developments has also severe limitations. For instance, even though they are biodegradable, their slow dissolution rates and sturdiness make them relatively easy to accumulate within tissues and during trafficking. Indeed, if inadequate surface modifications are done, these particles show a certain tendency to aggregate and therefore produce thrombosis and embolisms. On the other hand, the delivery of functional biomacromolecules with silica-based transporters presents important limitations too, the main one being the possible immune response of patients. In fact, many nanomedical designs fail when they are tested in immunocompetent models. Fortunately, the technology needed to overcome these problems is known and although the use of biopolymers such as nucleotides, proteins and peptides can alleviate these side effects to a large extent, this technological development is still in its infancy.

In addition to previous issues, the development of oligonucleotide drug delivery technologies also led to other limitations: the use of positively charged supports and polymers. This reduces overall biocompatibility in comparison with neutral or anionic systems due to the higher hemolysis rates. Although some research groups have tried to solve this problem by using fewer positive charges or by masking polycations, the truth is that it seems to be a rather difficult problem to solve. Proof of this is that all new-generation COVID-19 vaccines have been developed using liposome technologies which is completely different from solutions offered by oligonucleotide delivery with silica nanoparticles. Similarly, in the case of protein and enzyme delivery, there also are important limitations to be solved. In the case of in-pore loading, it is necessary to adapt pore sizes to proteins sizes and shapes while on surface location technologies that avoid the formations of protein coronas and avoid the blockage of active sites remain unsolved.

Regardless of these drawbacks, mesoporous silica nanotechnology also offers significant advantages. Among them, the main advance is the ease of developing combined therapies, which to our opinion are the future of anticancer therapies, as they have proved to be the most successful strategies to achieve complete remission of tumors. Moreover, and as outlined in this revision, silica offers also interesting possibilities for the development of topical and oral therapeutics which do not require systemic injections and may favor translation into clinical practice. Even though silica nanotechnology does not reach clinical practice, we still foresee a brilliant future for this technology. To our opinion, silica is already established as an important think tank for future nanomedical technologies and will still be in a near future.

Reviewer 2 Report

In this work authors have covered relevant advances based on therapeutic biomacromolecules loaded-mesoporous silica particles and their application as nanomedicines. The authors are optimistic about the future of mesoporous silica nanomaterials, particularly in terms of the controlled delivery of nucleic acid and protein-based therapeutics.

Author Response

In this work authors have covered relevant advances based on therapeutic biomacromolecules loaded-mesoporous silica particles and their application as nanomedicines. The authors are optimistic about the future of mesoporous silica nanomaterials, particularly in terms of the controlled delivery of nucleic acid and protein-based therapeutics.

Despite we do believe silica-based nanotechnology has the potential to become a powerful tool to develop new generation nanomedicines, we also understand reviewer’s point of view. To be more critical with the actual state of the art of silica, we have rewritten the conclusion section in accordance. We hope this new text will please you. Additionally, to improve readability, we have submitted the manuscript to an English editing. You can find the track changes version below.

As mentioned throughout the manuscript, the capacity and versatility of nanometric porous silica materials to develop potential therapeutic systems is directly enormous. This is partially consequence of its feasibility to develop novel strategies with interest for nanomedical approaches. Moreover, silica offers one of the simplest methods to develop nanosystems for combination therapies; which is clearly demonstrated by the great number of systems with therapeutic potential that have been reported up to date. In addition to the delivery of oligonucleotides, peptides and proteins reviewed herein, silica particles also offer many other possibilities to create nanotherapeutics such as targeted nanomedicines and stimuli responsive systems. But also, and consequence of their easy tunable porous structure, it permits to house within particles’ matrices a important number of small and medium-sized molecules of practically any chemical nature known.

However, the use of mesoporous silica for nanomedical developments has also severe limitations. For instance, even though they are biodegradable, their slow dissolution rates and sturdiness make them relatively easy to accumulate within tissues and during trafficking. Indeed, if inadequate surface modifications are done, these particles show a certain tendency to aggregate and therefore produce thrombosis and embolisms. On the other hand, the delivery of functional biomacromolecules with silica-based transporters presents important limitations too, the main one being the possible immune response of patients. In fact, many nanomedical designs fail when they are tested in immunocompetent models. Fortunately, the technology needed to overcome these problems is known and although the use of biopolymers such as nucleotides, proteins and peptides can alleviate these side effects to a large extent, this technological development is still in its infancy.

In addition to previous issues, the development of oligonucleotide drug delivery technologies also led to other limitations: the use of positively charged supports and polymers. This reduces overall biocompatibility in comparison with neutral or anionic systems due to the higher hemolysis rates. Although some research groups have tried to solve this problem by using fewer positive charges or by masking polycations, the truth is that it seems to be a rather difficult problem to solve. Proof of this is that all new-generation COVID-19 vaccines have been developed using liposome technologies which is completely different from solutions offered by oligonucleotide delivery with silica nanoparticles. Similarly, in the case of protein and enzyme delivery, there also are important limitations to be solved. In the case of in-pore loading, it is necessary to adapt pore sizes to proteins sizes and shapes while on surface location technologies that avoid the formations of protein coronas and avoid the blockage of active sites remain unsolved.

Regardless of these drawbacks, mesoporous silica nanotechnology also offers significant advantages. Among them, the main advance is the ease of developing combined therapies, which to our opinion are the future of anticancer therapies, as they have proved to be the most successful strategies to achieve complete remission of tumors. Moreover, and as outlined in this revision, silica offers also interesting possibilities for the development of topical and oral therapeutics which do not require systemic injections and may favor translation into clinical practice. Even though silica nanotechnology does not reach clinical practice, we still foresee a brilliant future for this technology. To our opinion, silica is already established as an important think tank for future nanomedical technologies and will still be in a near future.

Reviewer 3 Report

In the present manuscript entitled “Delivery of Therapeutic Biopolymers employing Silica-based Nanosystems” the authors have compiled different reports in which the delivery of different biopolymers is assisted by silica-based nanosystems. Although the present submission is exhaustive yet English is poor in many places. Authors are advised to check the manuscript carefully. Ongoing clinical trials and the clinical status of silica-based nanosystems for biopolymers are missing which must be included in the manuscript. The authors' critical comments are missing in the entire manuscript which should be included. Future directions should also be included. The images should also be improved. Some more images may be included to improve the overall quality of the manuscript.

Author Response

In the present manuscript entitled “Delivery of Therapeutic Biopolymers employing Silica-based Nanosystems” the authors have compiled different reports in which the delivery of different biopolymers is assisted by silica-based nanosystems. Although the present submission is exhaustive yet English is poor in many places. Authors are advised to check the manuscript carefully. Ongoing clinical trials and the clinical status of silica-based nanosystems for biopolymers are missing which must be included in the manuscript. The authors' critical comments are missing in the entire manuscript which should be included. Future directions should also be included. The images should also be improved. Some more images may be included to improve the overall quality of the manuscript.

We appreciate all comments made, and as suggested we have included critical comments and future directions in the conclusion section.

As mentioned throughout the manuscript, the capacity and versatility of nanometric porous silica materials to develop potential therapeutic systems is directly enormous. This is partially consequence of its feasibility to develop novel strategies with interest for nanomedical approaches. Moreover, silica offers one of the simplest methods to develop nanosystems for combination therapies; which is clearly demonstrated by the great number of systems with therapeutic potential that have been reported up to date. In addition to the delivery of oligonucleotides, peptides and proteins reviewed herein, silica particles also offer many other possibilities to create nanotherapeutics such as targeted nanomedicines and stimuli responsive systems. But also, and consequence of their easy tunable porous structure, it permits to house within particles’ matrices a important number of small and medium-sized molecules of practically any chemical nature known.

However, the use of mesoporous silica for nanomedical developments has also severe limitations. For instance, even though they are biodegradable, their slow dissolution rates and sturdiness make them relatively easy to accumulate within tissues and during trafficking. Indeed, if inadequate surface modifications are done, these particles show a certain tendency to aggregate and therefore produce thrombosis and embolisms. On the other hand, the delivery of functional biomacromolecules with silica-based transporters presents important limitations too, the main one being the possible immune response of patients. In fact, many nanomedical designs fail when they are tested in immunocompetent models. Fortunately, the technology needed to overcome these problems is known and although the use of biopolymers such as nucleotides, proteins and peptides can alleviate these side effects to a large extent, this technological development is still in its infancy.

In addition to previous issues, the development of oligonucleotide drug delivery technologies also led to other limitations: the use of positively charged supports and polymers. This reduces overall biocompatibility in comparison with neutral or anionic systems due to the higher hemolysis rates. Although some research groups have tried to solve this problem by using fewer positive charges or by masking polycations, the truth is that it seems to be a rather difficult problem to solve. Proof of this is that all new-generation COVID-19 vaccines have been developed using liposome technologies which is completely different from solutions offered by oligonucleotide delivery with silica nanoparticles. Similarly, in the case of protein and enzyme delivery, there also are important limitations to be solved. In the case of in-pore loading, it is necessary to adapt pore sizes to proteins sizes and shapes while on surface location technologies that avoid the formations of protein coronas and avoid the blockage of active sites remain unsolved.

Regardless of these drawbacks, mesoporous silica nanotechnology also offers significant advantages. Among them, the main advance is the ease of developing combined therapies, which to our opinion are the future of anticancer therapies, as they have proved to be the most successful strategies to achieve complete remission of tumors. Moreover, and as outlined in this revision, silica offers also interesting possibilities for the development of topical and oral therapeutics which do not require systemic injections and may favor translation into clinical practice. Even though silica nanotechnology does not reach clinical practice, we still foresee a brilliant future for this technology. To our opinion, silica is already established as an important think tank for future nanomedical technologies and will still be in a near future.

Moreover, as also pointed out, the text has gone through language edition. We hope all changes made will improve readability and overall quality. You can find the track changes version below.

Regarding figures, we agreed to maintain the number but with a significant increase in the information given. We believe there is much variability from one nanosystem to another to be summarized in figures rather than in the two comprehensive tables provided.

Round 2

Reviewer 1 Report

Accepted

Reviewer 3 Report

Responses are satisfactory.